# Parameter variations in personalized electrophysiological models of human heart ventricles

**Konstantin Ushenin**[1,2]*, **Vitaly Kalinin**[4], **Sukaynat Gitinova**[3], **Oleg Sopov**[3], **Olga Solovyova**[1,2]

**1** Institute of Natural Sciences and Mathematics, Ural Federal University, Ekaterinburg, Russia, **2** Institute of Immunology and Physiology of the Ural Branch of the RAS, Ekaterinburg, Russia, **3** Department of Surgical Treatment of Tachyarrhythmias, A.N. Bakulev National Medical Research Center of Cardiovascular Surgery, Moscow, Russia, **4** EP Solution SA, Yverdon-les-bains, Switzerland

\* konstantin.ushenin@urfu.ru

**Data Availability Statement:** Relevant data are within the paper and its Supporting information files. Raw data recorded from patients cannot be shared publicly because of national laws and Ethics Committee protocol. Data are available from the A.

## Abstract

The objectives of this study were to evaluate the accuracy of personalized numerical simulations of the electrical activity in human ventricles by comparing simulated electrocardiograms (ECGs) with real patients' ECGs and analyzing the sensitivity of the model output to variations in the model parameters. We used standard 12-lead ECGs and up to 224 unipolar body-surface ECGs to record three patients with cardiac resynchronization therapy devices and three patients with focal ventricular tachycardia. Patient-tailored geometrical models of the ventricles, atria, large vessels, liver, and spine were created using computed tomography data. Ten cases of focal ventricular activation were simulated using the bidomain model and the TNNP 2006 cellular model. The population-based values of electrical conductivities and other model parameters were used for accuracy analysis, and their variations were used for sensitivity analysis. The mean correlation coefficient between the simulated and real ECGs varied significantly (from $r = 0.29$ to $r = 0.86$) among the simulated cases. A strong mean correlation ($r > 0.7$) was found in eight of the ten model cases. The accuracy of the ECG simulation varied widely in the same patient depending on the localization of the excitation origin. The sensitivity analysis revealed that variations in the anisotropy ratio, blood conductivity, and cellular apicobasal heterogeneity had the strongest influence on transmembrane potential, while variation in lung conductivity had the greatest influence on body-surface ECGs. Futhermore, the anisotropy ratio predominantly affected the latest activation time and repolarization time dispersion, while the cellular apicobasal heterogeneity mainly affected the dispersion of action potential duration, and variation in lung conductivity mainly led to changes in the amplitudes of ECGs and cardiac electrograms. We also found that the effects of certain parameter variations had specific regional patterns on the cardiac and body surfaces. These observations are useful for further developing personalized cardiac models.

N. Bakulev National Medical Research Center of Cardiovascular Surgery for researchers who meet the criteria for access to confidential data (contact via info@bakulev.ru).

**Funding:** The development of personalized computer models was performed as part of the project that is supported by the Russian Science Foundation (https://rscf.ru/en/) in the form of a grant awarded to OS (19-14-00134). Computational resources and software development were covered by government assignment for (1) the Institute of Immunology and Physiology, Ural Branch of the Russian Academy of Sciences (https://iip.uran.ru/) in the form of a salary for OS and KU (AAA-A21-121012090093-0) and (2) Ural Federal University (https://urfu.ru/) in the form of a grant awarded to OS (02. A03.21.0006). EP Solutions SA, Yverdon-les-Bains, Switzerland (https://ep-solutions.ch) provided support in the form of a salary for VK and consultancy fees and travel grants awarded to KU. The specific roles of these authors are articulated in the "Author Contributions" section. The funders had no further role in study design, data collection and analysis, decision to publish, or preparation of the manuscript.

**Competing interests:** The authors have read the journal's policy and have the following competing interests: VK is a shareholder and employee of EP Solutions SA, Yverdon-les-Bains, Switzerland (https://ep-solutions.ch). KU had a consultancy agreement with and received travel grants from EP Solutions SA. This does not alter our adherence to PLOS ONE policies on sharing data and materials. There are no patents, products in development or marketed products associated with this research to declare.

## Introduction

The mathematical and numerical modeling of cardiac electrical activity in humans is of considerable significance in basic and clinical cardiac electrophysiology. State-of-the-art mathematical models, including the bidomain model of myocardial tissue, coupled with biophysically detailed cellular ionic models, can provide a physiologically-adequate simulation of electrical activity in the myocardium [1]. Cardiac imaging techniques, such as computed tomography (CT) and magnetic resonance imaging (MRI), enable models to include the personal anatomy of the heart, torso, and internal organs. These methods bring researchers closer to solving the challenging problem of creating personal models of the human heart electrical activity. Successful examples of using personalized cardiac models in clinical practice, particularly to predict vulnerability to life-threatening arrhythmia and plan optimal personalized therapy, have been reported [2–8]. However, creating patient-specific cardiac models as a routine clinical procedure is still far from a complete solution.

An important challenge facing the construction of personalized cardiac models is the lack of personalized information on the biophysical parameters used in the model equations. Model parameters, such as the intracellular and extracellular conductivity of the myocardial tissue, membrane capacity, surface-to-volume ratio, parameters of the ionic currents in cardiomyocytes, and electrical conductivity of internal organs in the chest, can vary significantly among individuals [9]. Moreover, myocardial tissue is heterogeneous, and the spatial distribution of its cellular and tissue properties can have an idiosyncratic pattern, particularly after cardiac remodeling in pathological conditions. Thus, complete personalized information on all model parameters is not available, forcing most biophysical cardiac models to be based on average population values for parameters extracted from a few research reports, and such data are not always consistent.

For this reason, developing methods to validate and individually refine the models using actual measurements of cardiac electrical activity in individuals is important, and the chief task is to assess the sensitivity of model output to variations in model parameters. Such a sensitivity analysis would rank the model parameters according to their impacts on model outputs.

Several recent studies have focused on validating and individually adjusting models of cardiac electrical activity. Undoubtedly, the most informative data for validating cardiac electrophysiological models can be derived from invasive cardiac mapping data [10]. However, invasive cardiac mapping has limited indications for patients. The more practically appropriate method for model parameterization is using data from ECG body surface mapping (e.g., multi-channel recording of ECG signals from the human body surface). Several recent articles have employed such approaches [9–13], most of which used standard 12-lead ECGs. Only a few articles [9, 11] have reported using more informative multi-channel body-surface mapping (BSM) in addition to 12-lead ECGs.

In these works, the electrical activity of the heart in the sinus rhythm was modeled and validated accordingly. Simulated ECGs have significant sensitivity to variations in the initial conditions of ventricular excitation [14], so to simulate ECGs of patients in sinus rhythm with normal ventricular conduction, information on the His-Purkinje system must be accounted for in the ECG simulation pipeline to define the appropriate initial conditions of ventricular excitation. The His bundle branches and Purkinje fiber network have significant variability [15–17], which leads to significant variability in the earliest ventricular activation [18, 19]. However, in vivo methods for identifying the personal structure of the cardiac conduction system have not been developed, so a lack of patient-specific information about the His-Purkinje system can cause inaccuracy in models of ventricle activation and patient-specific ECGs.

In a recent study [20], researchers compared simulated ECGs produced by a human ventricular model with excitation originating from the focal origins with actual ECG records. This approach allowed them to avoid simulating His-Purkinje conduction. Anatomical data and ECG records of patients with implanted pacemakers and patients with focal ventricular tachycardia were used. However, they employed a simplified mathematical model of cardiac electrical activity: They simulated the electrical activation of ventricles using the monodomain version of Mitchell-Schaeffer's phenomenological two-variable model and computed ECGs using the current dipole approach. Thus, their results must be re-examined using more realistic mathematical models.

Several recent works have also analyzed the sensitivity of model output to variations in model parameters. For example, in [21], the authors focused on exploring the morphology of simulated 12-lead ECGs to evaluate model assumptions. They tested various model features, such as the bidomain and monodomain versions of myocardial homogenization, heart-torso coupling, normal and pathological His-Purkinje conduction, myocardial heterogeneity and anisotropy, electrical conductivity of the torso, and effects of the capacity and resistance of the pericardium. In all cases, the simplified Mitchell-Schaeffer's model was used as the cellular model of the ventricle myocardium. In subsequent work on this issue, a realistic ionic TNNP 2006 cellular model for human ventricular tissue [22] was utilized. Using this cellular model, Keller et al. examined the effects of varying tissue conductivity on simulated ECGs [23]. In [9, 24], the authors evaluated the effects of ventricular wall deformations and cellular $I_{Ks}$ heterogeneity on the T-wave morphology of simulated ECGs. Subsequently, Sanchez et al. [10] investigated the sensitivity of ECGs and the left ventricular activation sequence to changes in 39 parameters of the ventricular electrophysiological model. Note that in these works, ECGs and ventricular activation [10] were subjected to sensitivity analyses, while the sensitivity of underlying transmembrane potentials on the heart surface, and electrical potentials on the heart and body surface were not discussed. Moreover, each of the aforementioned studies used only one model of ventricular geometry despite ventricular anatomy varying significantly between subjects affecting electrical activity [25]. Furthermore, these works mainly investigated the effect of parameter variation on the numerical value of the integral errors in model output, but other phenomena are also worth investigating, such as the analysis of modeling errors in terms of its spatial distribution on the surface of the torso and heart and the influence of parameter variation on the temporal and morphological characteristics of cardiac electrical signals. These issues have not been explored in sufficient detail.

The present study is devoted to validating and individually fitting models of human cardiac ventricular electrical activity. We focused on a few insufficiently researched points, as noted above. First, we created personalized anatomical models of the heart-torso to determine the level of ECG simulation accuracy that can be achieved with the bidomain model, in combination with a realistic ionic model for human ventricular cardiomyocytes that uses average, population-based values for the model parameters. As in [20], we limited ourselves to modeling ventricular excitation arising from focal sources to avoid inaccuracy in the initial conditions associated with His-Purkinje conduction. However, unlike [20], we used the TNNP 2006 model as a physiologically accurate ionic model and the bidomain model with a bath as the most physically realistic method of ECG simulation.

The second objective of this study is to evaluate the effects of variation in model parameters within the reported physiological ranges on model outputs. In addition to a sensitivity analysis of the 12-lead ECG and BSM electrode array, we also analyzed the effects of parameter variation on transmembrane potentials in myocardial tissue and on extracellular potentials. Especially, we focused our attention on the regional and local sensitivity.

## Methods

### Patient data

Clinical data from six patients (4 male, 2 female, age 48.8±17.6) who had been examined at the Bakoulev center for cardiovascular surgery (Moscow, Russia) were used in the study. Table 1 summarizes the baseline characteristics of the patients.

Three patients had hypertrophic (P1), dilated (P2), and arrhythmogenic (P3) cardiomyopathy and class II heart failure according to the classification of the New York Heart Association (NYHA class). These patients had implanted devices for cardiac resynchronization therapy (CRT). Pacing electrodes of the CRT devices were installed in a generally accepted manner. Right-ventricle (RV) pacing electrodes were positioned in the apex of the RV in all cases; left-ventricle (LV) pacing electrodes were introduced through the coronary sinus to the epicardial surface of the lateral wall of the LV and were installed in the superior-lateral vein (n = 2) and lateral vein (n = 1). Table 1 presents the data on patients with CRT devices.

Three other patients (P4, P5, and P6) had focal ventricular tachycardia. In one case (P5), focal activity originated from the myocardium diverticula in the apical area of the LV. The other two patients (P4 and P6) with structurally healthy hearts had idiopathic ventricular tachycardia with ectopic foci in the right ventricular outflow tract. One had an additional ectopic focus in the lateral wall of the RV. The localization of ectopic foci was detected by electroanatomical cardiac mapping with a CARTO 3 system (Biosense Webster Inc., Diamond Bar, USA) and confirmed by the successful result of cryoablation surgery (P5) or radiofrequency catheter ablation (P4, P6), as well as the results of six-month follow-ups. In patients with CRT devices, BSM was performed under RV and LV pacing during CRT device programming. In the other patients, BSM was conducted during a focal ventricular tachycardia rhythm, prior to performing interventional procedures. The experimental protocol was approved by the Ethics Committee of the A. N. Bakoulev National Medical Research Center of Cardiovascular Surgery (Protocol No. 2, 30.05.2017). All patients provided written informed consent for the CT and electrophysiological procedures and agreed to data retrieval and analysis.

### Data preprocessing

Body-surface electrode positioning was detected by an expert using CT data and Amycard 01 K software [26]. Body-surface ECG recordings were analyzed by an expert. A few BSM leads with strong non-eliminating noise due to poor connection to the skin were excluded. Between 210 and 224 electrodes were used in the following analysis. The heart and torso were segmented from CT data with Amycard 01 software by an expert. Biventricular 3D models were created in the end-diastolic phase of the cardiac cycle using ECG-gated CT data. To create

**Table 1. Baseline characteristics of patients (P1-P6) enrolled in the study.**

| N | Activation in models | Gender | Age | Diagnosis | Origin of ventricular ectopic activity |
|---|---|---|---|---|---|
| P1 | LV, RV | F | 67 | HCMP, LBBB, HF of II class (NYHA) | CRT device, RV, LV, and biventricular pacing |
| P2 | LV, RV | M | 66 | DCMP, LBBB, HF of II class (NYHA) | CRT device, RV, LV, and biventricular pacing |
| P3 | LV, RV | M | 56 | ACMP, LBBB, HF of II class (NYHA) | CRT device, RV, LV, and biventricular pacing |
| P4 | RV1, RV2 | M | 35 | Idiopathic focal ventricular tachycardia | Right ventricle outflow tract, Right ventricle lateral wall |
| P5 | LV | M | 46 | LV apical diverticulum | Diverticulum in the apical region of the LV |
| P6 | RV | F | 23 | Idiopathic focal ventricular tachycardia | Right ventricle outflow tract |

Abbreviations: hypertrophic cardiomyopathy (HCMP), left ventricular bundle branch block (LBBB), heart failure (HF), New York Heart Association (NYHA) class, arrhythmogenic cardiomyopathy (ACMP), and cardiac resynchronization therapy (CRT).

volume-conductor models with piecewise homogeneous electrical conductivity, the ventricles, atria, large vessels, liver, and spine were also segmented from CT data by an expert. The surface triangle meshes of the heart and torso, as well as the internal organs, were subsequently built with proprietary EP Solution SA software. Thereafter, 3D tetrahedral meshes for finite element simulation were generated using GMSH open-source software. Meshes were improved iteratively using the refine-by-splitting algorithm [27].

Sites of ventricular excitation were located in the following manner. In patients with CRT devices (P1–P3), positions of stimulation sites were defined as the positions of an implanted electrodes' pacing poles on CT data. In patients with focal ventricular tachycardia (P4–P6), the origins of ventricular excitation were found using invasive cardiac mapping with a CARTO 3 system. To translate their localization to the CT-based cardiac models, CARTO-based 3D electroanatomical models of ventricles were merged with CT data using a previously described method [26].

Thus, a dataset was prepared that included body-surface and 12-lead ECGs, finite element meshes of the heart, torso, and internal organs, and clinically defined positions of activation origins for ten ventricular-activation cases in six patients.

## Models of cardiac electrical activity and passive electrical properties of heart and human body

Denote $\Omega$ as the myocardial domain, $\Omega_b$ as the torso domain, $\partial\Omega$ as a boundary of the myocardium, and $\partial\Omega_b$ as a boundary of the torso. We assumed that $\Omega_b$ is a passive volume conductor without electrical sources and that $\Omega$ is an excitable medium. We used a bidomain model with bath and full coupling formulation of the boundary conditions to simulate cardiac electrical activity:

$$\begin{cases} \nabla \cdot (\Sigma_i(\nabla V_m + \nabla\phi_e)) = \beta(C_m\frac{\partial V_m}{\partial t} + I_{\mathrm{ion}} + I_{\mathrm{app}}), \ \mathrm{in} \ \Omega \times (0, T] \\ \nabla \cdot ((\Sigma_i + \Sigma_e)\nabla\phi_e) = -\nabla \cdot (\Sigma_i\nabla V_m), \ \mathrm{in} \ \Omega \times (0, T] \\ \nabla \cdot (\Sigma_b\nabla\phi_e) = 0, \ \mathrm{in} \ \Omega_b \times (0, T], \end{cases} \quad (1)$$

$$\phi_e = \phi_b \ \mathrm{on} \ \partial\Omega \times (0, T], \quad (2)$$

$$\mathbf{n} \cdot (\Sigma_b\nabla\phi_b) - \mathbf{n} \cdot (\Sigma_e\nabla\phi_e) = 0 \ \mathrm{on} \ \partial\Omega \times (0, T], \quad (3)$$

$$\mathbf{n} \cdot (\Sigma_i\nabla V_m) + \mathbf{n} \cdot (\Sigma_i\nabla\phi_e) = 0 \ \mathrm{on} \ \partial\Omega \times (0, T], \quad (4)$$

$$\mathbf{n} \cdot (\Sigma_b\nabla\phi_b) = 0 \ \mathrm{on} \ \partial\Omega_b \times (0, T]. \quad (5)$$

where $\phi_e$ is extracellular potential, $\phi_i$ is intracellular potential, $V_m = \phi_i - \phi_e$ is transmembrane potential, $\phi_b$ is electrical potential on the torso, $C_m$ is membrane capacitance, $\beta$ is surface-to-volume ratio, $I_{ion}$ and $I_{app}$ are ionic and stimulation currents, respectively $\Sigma_b = \mathrm{diag}(\sigma_b, \sigma_b, \sigma_b)$ is torso conductivity, and $\Sigma_e$ and $\Sigma_i$ are extracellular and intracellular conductivity tensors, respectively. We assume the torso to be an isotropic volume conductor and the myocardium to be an anisotropic volume conductor. Moreover, electrical potentials on the body surface were computed subject to an additional condition of Wilson's central terminal signal: equality to zero. The anisotropic electrical conductivity of intracellular and extracellular media was introduced by assigning electrical conductivity tensors $\Sigma_i$ and $\Sigma_e$, respectively, to each node of the tetrahedral mesh. Following [1, 21, 23], we assume equal conductivities transversal to the

main direction of the fiber-orientation vector. With this assumption, tensors $\Sigma_i$ and $\Sigma_e$ can be represented as follows:

$$\sum\nolimits_i = M \begin{pmatrix} \sigma_{li} & 0 & 0 \\ 0 & \sigma_{ti} & 0 \\ 0 & 0 & \sigma_{ti} \end{pmatrix} M^T, \sum\nolimits_e = M \begin{pmatrix} \sigma_{le} & 0 & 0 \\ 0 & \sigma_{te} & 0 \\ 0 & 0 & \sigma_{te} \end{pmatrix} M^T,$$

where matrix $M$ is a rotational basis that is determined by the fiber orientation and $\sigma_{li}$, $\sigma_{ti}$ and $\sigma_{le}$, $\sigma_{te}$ are intracellular and extracellular conductivities along and across the fiber, respectively.

We used Roth's mathematical framework [28] to assign values to the parameters $\sigma_{li}$, $\sigma_{ti}$ and $\sigma_{le}$, $\sigma_{te}$. According to this approach, they are calculated using the following formulas:

$$\sigma_{li} = \sigma, \tag{6}$$

$$\sigma_{ti} = \sigma \left( 1/\frac{\lambda_L}{\lambda_T} \right)^2 \left( \frac{1 + \alpha(1 - \varepsilon)}{1 + \alpha} \right), \tag{7}$$

$$\sigma_{le} = \sigma \frac{1}{\alpha}, \tag{8}$$

$$\sigma_{te} = \sigma \left( 1/\frac{\lambda_L}{\lambda_T} \right)^2 \left( \frac{1 + \alpha(1 - \varepsilon)}{1 + \alpha} \right) \frac{1}{\alpha(1 - \varepsilon)} \tag{9}$$

where $\sigma$ is basic myocardial conductivity, and $\frac{\lambda_L}{\lambda_T}$ is the so-called anisotropy ratio coefficient, $\alpha = \frac{\sigma_{li}}{\sigma_{le}}$, $\varepsilon = 1 - \frac{\sigma_{le}/\sigma_{te}}{\sigma_{li}/\sigma_{ti}}$. Following [21], we set these values as $\sigma = 3$, $\alpha = 1$, and $\varepsilon = 0.75$. Therefore, the values of $\sigma_{li}$, $\sigma_{ti}$ and $\sigma_{le}$, $\sigma_{te}$ are governed by a single parameter: the anisotropy ratio coefficient $\frac{\lambda_L}{\lambda_T}$. The rotation basis M was calculated using myocardial fiber vectors, which were determined in the myocardium volume by a rule-based approach (see [29] for details).

We employed the TNNP 2006 cellular model for human ventricle cardiomyocytes [22] to compute the transmembrane ionic current $I_{ion}$. The cellular model has three proposed versions: epicardial cardiomyocytes, endocardial cardiomyocytes, and hypothetical transmural M-cells; in this study, we used the epicardial and endocardial versions of the model. Cellular transmural heterogeneity was introduced discretely by dividing the ventricular walls into two layers with either epicardial (epi) or endocardial (endo) cell types. The epicardial and endocardial versions of the TNNP 2006 model were utilized to simulate electrical activity in the epicardial and endocardial layers, respectively. The epi/endo heterogeneity coefficient $H_{TR} \in [0, 1]$ determined the fraction of the transmural depth occupied by these two cellular layers (0: endo-type cells only; 0.5: half-and-half epi/endo-type cells; 1: epi-type cells only).

Similar to [9], cellular apicobasal heterogeneity was introduced by the linear dependence of the conductivity parameter $g_{Ks}$ for the slow potassium current $I_{Ks}$ on the coordinate on the longitudinal ventricular axis from the apex to the base: $g_{Ks} = (0.392 - 0.294H_{AB})$, where $H_{AB} \in [0, 1]$ is a variable parameter. We assumed a physiologically realistic range for $H_{AB} \in [0.75, 1]$.

Consequently, in our model, the anisotropic electrical conductivity of the myocardium was governed by the anisotropic ratio coefficient $\frac{\lambda_L}{\lambda_T}$; cellular transmural heterogeneity was governed by the thickness ratio of the epicardial and endocardial layers of the ventricles $H_{TR}$, and cellular apicobasal heterogeneity was governed by the parameter $H_{AB}$.

In this study, we assumed torso electrical conductivity to be isotropic. Torso-conductivity heterogeneities were introduced by the following simplified approach. Anatomical structures with identical electrical conductivities were joined to obtain larger regions with homogeneous conduction properties: the cardiac biventricular region, the lung region (including both the left and right lungs), the blood region (including blood in the ventricular and atrial cavities, the aorta, and the pulmonary veins), the spine region, and the liver region. Finally, electrical conductivity values were assigned to the mesh elements according to their locations in the specific regions.

## Fixed and varied parameters of models

Parameters of the models can be divided into three groups. The first group consists of parameters of the cellular model, such as ionic currents in cardiomyocytes. The second group includes parameters of the bidomain model: cell membrane capacity, the surface-to-volume ratio of cardiomyocytes, and coefficients of the conductivity tensors of myocardial tissue. The third group of model parameters includes parameters of the torso organ conductivities. In all our models, parameter values of the first group, except for the potassium current, were taken from an original work [22] since we assume that the TNNP 2006 model of human ventricular cardiomyocytes has an optimal balance between model complexity and requirements in computational power.

Moreover, for all simulation cases, we used the same values for the following parameters: membrane conductivity ($C_m$), surface-to-volume ratio ($\beta$), basic conductivity of the torso ($\sigma_b$), basic conductivity of the myocardium ($\sigma$), and Roth's mathematical framework parameters ($\alpha$, $\varepsilon$). These parameter values were taken from published works [10, 11, 21–24, 28]. Table 2 shows these values.

To analyze the sensitivity of model output to changes in its parameters, we varied the following: coefficients of anisotropic electrical conductivity of the myocardium ($\sigma_{li}$, $\sigma_{le}$, $\sigma_{ti}$, $\sigma_{te}$) and values of electrical conductivity of the lungs ($\sigma_{lungs}$), liver ($\sigma_{liver}$), and spine ($\sigma_{spine}$), as well as blood in the heart chambers and large vessels ($\sigma_{blood}$). Moreover, we varied the parameters of apicobasal and transmural heterogeneity of potassium currents ($g_{Ks}$, $g_{to}$). An example of the significant influence these parameters have shown in a work [9].

We used a special approach when varying the parameters of anisotropic electrical conductivity of the myocardium ($\sigma_{li}$, $\sigma_{le}$, $\sigma_{ti}$, $\sigma_{te}$). Since we calculated the values of myocardial conductivity parameters ($\sigma_{li}$, $\sigma_{le}$, $\sigma_{ti}$, $\sigma_{te}$) using the Ross framework, variations in the anisotropy ratio ($\lambda_L/\lambda_T$) led to corresponding changes in these values. This approach allowed us to vary only the value of the anisotropy coefficient ($\lambda_L/\lambda_T$) instead of alternating changes of values of four parameters in our sensitivity analysis. The list of parameters whose values we have varied is given in Table 3. The same table shows the physiological ranges of their values with links to the corresponding works.

**Table 2. Model parameters that were not varied in the experiments.**

| Physiological parameter | Notation | Value | Unit | Literature |
|---|---|---|---|---|
| Membrane capacitance | $C_m$ | 1 | $\mu F/cm^2$ | [22] |
| Surface-to-volume ratio | $\beta$ | 400 | $cm^{-1}$ | [10] |
| Myocardial conductivity (basic) | $\sigma$ | 3.0 | $mS/cm$ | [10, 11, 21, 28], |
| Roth's framework parameter $\alpha$ | $\alpha$ | 1 | | [28] |
| Roth's framework parameter $\varepsilon$ | $\varepsilon$ | 0.75 | | [28] |
| Torso conductivity (basic) | $\sigma_b$ | 2.0 | $mS/cm$ | [23], |

**Table 3. Model parameters that varied in the simulations.**

| Physiological parameters | Notation | Reference model value | Unit | Study variation range | Physiological variation range | Literature |
|---|---|---|---|---|---|---|
| Epi/endo heterogeniety coef. | $H_{TR}$ | 0.5 | | [0, 1] | [0.2, 0.6] | adapted from [9] |
| Apicobasal heterogeniety coef. | $H_{AB}$ | 1 | | [0, 1] | [0.7, 1.0] | adapted from [9] |
| Anisotropy ratio | $\lambda_L/\lambda_T$ | 2.5 | | [1.6, 6.0] | [2.0, 3.0] | [25, 31] |
| Lungs conductivity | $\sigma_{\text{lungs}}$ | 0.39 | $mS/cm$ | [0.39, 1.34] | [0.39, 1.34] | [9] |
| Blood conductivity | $\sigma_{\text{blood}}$ | 7 | $mS/cm$ | [4.35, 10] | [4.35, 10] | [9] |
| Spine conductivity | $\sigma_{\text{spine}}$ | 0.2 | $mS/cm$ | [0.05, 0.6] | [0.05, 0.6] | [9] |
| Liver conductivity | $\sigma_{\text{liver}}$ | 0.28 | $mS/cm$ | [0.28, 2.0] | [0.28, 2.0] | [9] |

The table shows reference values and the ranges of parameter variation with corresponding literature sources.

Among the variable parameter values, we selected a group of reference values. They are shown in the second column of Table 3. We used these reference values to compare the simulation results with the ECG of patients and as a reference point for the sensitivity analysis. For the reference model, the parameter values were assigned as follows. We set the parameter values $H_{TR}$, $H_{AB}$, and $\lambda_L/\lambda_T$ based on previous work [9], where these values are evaluated as the most physiologically correct. We used population-based values for parameters of the bidomain model based on previous works [9, 10, 21, 23], which carefully selected a plausible range of values to simulate adequate characteristics of ECGs recorded in patients. In particular, these parameter values allow the model to produce a realistic conduction velocity 0.5-0.6 m/s in myofiber direction and 0.15-0.25 m/s across the fibers, as reported in the previous work. Furthermore, these parameters provide a QRS width greater than 100 ms, which is close to patient recordings upon point stimulation. Finally, we used population-based values for the electrical conductivity of blood and the internal organs that has been reported in previous works [10, 23].

## Simulation of cardiac electrical activity

We performed excitation simulations of the ventricles originating from ectopic sources of precise patient-specific localization. For this purpose, pacing points in the geometrical ventricular models were placed on pacing and ectopic sites detected in the patients. Ventricular excitation was initiated by applying a rectangular impulse of stimulation current $I_{\text{stim}}$ (see Eq 1)) for 3 ms to a region with a radius of 3 mm while the initial conditions of the bidomain model had resting state values. Simulations of cardiac electrical activity were performed using Cardiac CHASTE software [30] on the supercomputer 'URAN' (Institute of Mathematics and Mechanics of Ural Branch of Russian Academy of Sciences). Simulation results included time-dependent values of the transmembrane potential and extracellular potentials (electrograms) in each node of the finite element mesh, electrical potential values (unipolar electrocardiograms) in each body-surface node of the finite element mesh, and both standard 12-lead ECGs and body-surface ECGs. The time resolution of the simulated signals was 1,000 frames per second.

## Model analysis

Here, we describe our approaches to compare model outputs with clinical data, outputs from models with varied parameters, and model sensitivity analysis. Let us denote a model output signal as **S**. This can be a simulated ECG signal produced by the model, which we compare with the ECG data measured for a patient denoted as **P**. The model signal **S** forms a set of value $\mathbf{S} = \{s_i^t | i \in I, t \in [0, T]\}$, where $t$ is the time from the $[0, T]$ interval, and $i$ is an index of

nodes in a subset *I* of model mesh nodes. The subset I may include all points of the finite element model, all points on certain surfaces, or a set of electrode tip locations on the body surface.

First, we compared experimental ECG data recorded in ten clinical cases in six patients with different ventricular activation protocols; we used simulations computed with a reference model with population-based parameters, which is common in modeling studies. Tables 2 and 3 show the reference values for some tissue-level model parameters, and other parameters were taken from an original article [22]. Then, we denote model signals computed for a patient case model with the reference parameter set as $R = \{r_i^t\}$. The distance between the simulated and patient ECG signals is denoted as

$$\Delta = \text{dist}(\mathbf{R}, \mathbf{P}) \tag{10}$$

First, we define $\Delta$ in terms of qualitative metrics using the correlation coefficient (CC) between the ECG signals for each body-surface electrode *i* separately:

$$\Delta_i = \text{CC}(\mathbf{R}_i, \mathbf{P}_i) = \frac{\sum_{t=0}^{T}(r_i^t - \bar{r}_i)(p_i^t - \bar{p}_i)}{\sqrt{\sum_{t=0}^{T}(r_i^t - \bar{r}_i)^2}\sqrt{\sum_{t=0}^{T}(p_i^t - \bar{p}_i)^2}}, \tag{11}$$

where $i \in I$, *I* is a set of body-surface electrodes; $\bar{r}_i$ and $\bar{p}_i$ are the mean values of signals.

Such metrics are conventional measures of the qualitative difference between signals and have been widely used in other studies [9, 11, 20, 23]. They are suitable for analyzing model errors on a BSM electrode array and allow one to reveal the spatial patterns of errors and regions of poor correlation between patient data and reference simulations.

The CC metrics are weakly sensitive to variations in the signal amplitudes, so we also calculated a normalized root mean square deviation (NRMSD) in each electrode *i* from the set *I* of body-surface electrodes:

$$\Delta_i = \text{NRMSD}(\mathbf{R}_i, \mathbf{P}_i) = \sqrt{\frac{\sum_{t=0}^{T}(r_i^t - p_i^t)^2}{T}} \frac{1}{\max_t p_i^t - \min_t p_i^t} \cdot 100\% \tag{12}$$

This metric accounts for the different ECG amplitudes recorded from different leads in the patient data.

Then, we choose a certain set of tissue-level model parameters (**X**; see Table 3) and analyze the dependence of the distance between the simulated and patient ECG on each parameter $x_{\text{var}} \in \mathbf{X}$:

$$\Delta(x_{\text{var}}) = \text{dist}(\mathbf{S}(x_{\text{var}}), \mathbf{P}) \tag{13}$$

Here, we use a relative Euclidean distance (RED) as an overall measure of the difference between the simulated and patient ECG signals in the entire set *I* of the BSM lead array:

$$\Delta(x_{\text{var}}) = \text{RED}(\mathbf{S}(x_{\text{var}}), \mathbf{P}) = \frac{\sqrt{\sum_{i=1}^{N}\sum_{t=0}^{T}(s_i^t(x_{\text{var}}) - p_i^t)^2}}{\sqrt{\sum_{i=1}^{N}\sum_{t=0}^{T}(p_i^t)^2}} \cdot 100\% \tag{14}$$

We addressed this task with the simplest possible analysis using one-by-one parameter variation in a physiologically non-implausible range, with other parameters fixed to the reference values. We computed the function $\Delta(x_{\text{var}})$ of each parameter $x_{\text{var}}$ at several tested values of $x_{\text{var}} \in [x_{min}, x_{max}]$ and then interpolated the function values on the entire parameter interval. We defined two ranges for each parameter variation: a physiological variation range (a physiologically non-implausible range) and a study variation range. The physiological variation range

corresponds to observed experimental values from real measurements reported in the literature (Table 3). Reference parameter values were taken from physiological ranges. The study variation range was widened beyond the physiological one for several parameters (i.e., the endo/epi coefficient, apicobasal heterogeneity coefficient, and anisotropy ratio) where the experimental data are less well defined.

In the third part of the model analysis, we used the RED metrics to analyze the model's sensitivity to parameter variation within the physiological range determined by the reference model outputs. In this case, a comparison was performed between signals from the reference model ($\mathbf{R}$) and models with different single parameters in the physiological range ($\mathbf{S}(x_{\mathrm{var}})$, $x_{\mathrm{var}} \in [x_{min}, x_{max}]$). Maximal RED metrics were used as a measure of model sensitivity within the physiological range of parameter variation:

$$\Delta_{x_{\mathrm{var}}} = \max_{x_{\mathrm{var}} \in [x_{\mathrm{min}}, x_{\mathrm{max}}]} \mathrm{RED}(\mathbf{S}(x_{\mathrm{var}}), \mathbf{R}) =$$

$$= \max_{x_{\mathrm{var}} \in [x_{\mathrm{min}}, x_{\mathrm{max}}]} \frac{\sqrt{\sum_{i=1}^{N} \sum_{t=0}^{T} \left(s_i^t(x_{\mathrm{var}}) - r_i^t\right)^2}}{\sqrt{\sum_{i=1}^{N} \sum_{t=0}^{T} \left(r_i^t\right)^2}} \cdot 100\% \tag{15}$$

This approach to sensitivity analysis is suitable for ranking model parameters with respect to their effects on model outputs. We used this measure of model sensitivity not only for ECG signals on the body surface but also for the transmembrane and extracellular potentials on myocardial surfaces and throughout the myocardial tissue. In the latter cases, we calculated $\Delta_{x_{\mathrm{var}}}$ on either the set $I$ of nodes from the surface or from the entire body of ventricles. We also used this approach to build sensitivity maps of the heart and torso surfaces to see the special effects of model parameter variation on different myocardial regions. In this case, we calculated $\Delta_{x_{\mathrm{var}}}$ in each node on the surfaces and analyzed the map patterns.

We employed a similar approach to analyze the effects of single-parameter variation on the physiologically significant characteristics of signals, such as ECG wave amplitudes, QRS width, action potential duration (APD)(Table 5). Each signal characteristic $U(\mathbf{S})$ is a scalar value, so we calculated the minimal and maximal relative value of the characteristic at different parameters with respect to the value in the reference model:

$$U_{\mathrm{min}} = \min_{x_{\mathrm{var}} \in [x_{\mathrm{min}}, x_{\mathrm{max}}]} U(\mathbf{S}(x_{\mathrm{var}}))/U(\mathbf{R}) \cdot 100\%, \tag{16}$$

$$U_{\mathrm{max}} = \max_{x_{\mathrm{var}} \in [x_{\mathrm{min}}, x_{\mathrm{max}}]} U(\mathbf{S}(x_{\mathrm{var}}))/U(\mathbf{R}) \cdot 100\%, \tag{17}$$

We adopted this approach from [9] for signal comparison and from [10] for physiological biomarker comparison.

## Results

Throughout this paper, the term reference simulation refers to each of the ten models computed with the reference parameters in Table 3. We compared patient electrocardiograms (PECGs) and simulated ECGs (SECGs) with reference parameters, and we compared the reference SECGs with SECGs computed with different model parameters.

### Comparison of reference simulations and patient ECG

In this section, we present results of the comparison of SECGs in each of the ten reference models and PECGs, which were recorded with BSM and standard 12-leads. Table 4 and Fig 1

**Table 4. Summary of correlation coefficient (CC) and NRMSD values for simulated and measured ECGs.**

| | Correlation (CC) | | | | | | | | NRMSD | | | | | | | |
| | Electrode vest | | | | 12 leads | | | | Electrode vest | | | | 12 leads | | | |
| | 25 perc. | 75 perc. | mean | sth | 25 perc. | 75 perc. | mean | sth | 25 perc. | 75 perc. | mean | sth | 25 perc. | 75 perc. | mean | sth |
|---|---|---|---|---|---|---|---|---|---|---|---|---|---|---|---|---|
| P1(LV) | 0.70 | 0.91 | 0.86 | 0.38 | 0.31 | 0.84 | 0.74 | 0.47 | 6.29% | 14.16% | 8.45% | 5.93% | 7.21% | 16.52% | 11.98% | 6.14% |
| P6(RV) | 0.66 | 0.93 | 0.86 | 0.31 | 0.81 | 0.92 | 0.87 | 0.10 | 4.41% | 8.93% | 5.69% | 6.01% | 3.50% | 6.68% | 4.47% | 2.25% |
| P3(LV) | 0.60 | 0.91 | 0.84 | 0.32 | 0.64 | 0.89 | 0.80 | 0.19 | 8.75% | 22.17% | 15.48% | 13.62% | 7.75% | 30.70% | 22.24% | 12.83% |
| P5(LV) | 0.57 | 0.87 | 0.83 | 0.36 | 0.81 | 0.88 | 0.87 | 0.18 | 10.29% | 18.76% | 13.75% | 9.14% | 6.04% | 11.62% | 6.63% | 5.60% |
| P4(RV1) | 0.40 | 0.91 | 0.83 | 0.51 | 0.87 | 0.91 | 0.91 | 0.22 | 9.86% | 31.61% | 17.25% | 23.08% | 8.14% | 17.60% | 13.98% | 10.82% |
| P1(RV) | 0.37 | 0.88 | 0.81 | 0.44 | 0.83 | 0.91 | 0.87 | 0.25 | 7.16% | 15.61% | 9.54% | 8.55% | 3.73% | 6.36% | 4.48% | 8.44% |
| P4(RV2) | 0.04 | 0.86 | 0.72 | 0.59 | -0.18 | 0.80 | 0.67 | 0.54 | 9.06% | 29.04% | 14.69% | 17.87% | 6.81% | 13.96% | 10.66% | 10.13% |
| P2(RV) | -0.08 | 0.90 | 0.77 | 0.61 | 0.69 | 0.88 | 0.87 | 0.62 | 8.93% | 19.95% | 12.56% | 10.61% | 5.29% | 9.07% | 6.93% | 9.11% |
| P2(LV) | -0.44 | 0.63 | 0.29 | 0.55 | -0.27 | 0.40 | 0.12 | 0.43 | 12.24% | 24.63% | 16.85% | 16.32% | 8.70% | 15.17% | 10.57% | 8.82% |
| P3(RV) | -0.48 | 0.91 | 0.69 | 0.73 | 0.66 | 0.95 | 0.89 | 0.31 | 10.83% | 31.92% | 18.06% | 26.45% | 9.10% | 17.62% | 13.06% | 18.08% |

Codes P1-P6 denote patients with an indication of the pacing ventricle (LV or RV).

summarize the results of the comparison in terms of the CC and NRMSD metrics (see formulas (12) and (13), where a set of nodes $I$ includes points of the 12 standard lead or electrodes from BSM).

The mean CC in BSM leads varies from 0.86 (cases P1(LV) and P6 (RV)) to 0.29 (P2(LV)), while the NRMSD varies from 5.69% (case P6(RV)) to 18.06% (case P3(RV)). Eight of the ten simulation cases (80%) have a mean CC higher than 0.7 (strong correlation [32]), and six (60%) have a mean NRMSD less than 15%. The models can be classified into three groups according to these mean CC and NRMSD values. The group with the highest accuracy includes three cases (P1(LV), P1(RV), and P6(RV)) that have mean CCs greater than 0.8 (very strong correlation [32]) and mean NRMSDs less than 10%. The group with low accuracy includes two cases (P2(LV) and P3 (RV)) with mean CCs less than 0.7 (low correlation [32]) and mean NRMSDs greater than 15%. The group with moderate accuracy includes the remaining five cases with mean CCs $\in$[0.7, 0.8] and mean NRMSDs $\in$[10%, 15%]. Despite most cases having rather strong mean correlations between the SECG and PECG values, most had highly variable CC values among BSM ECG leads in the same model. In particular, four

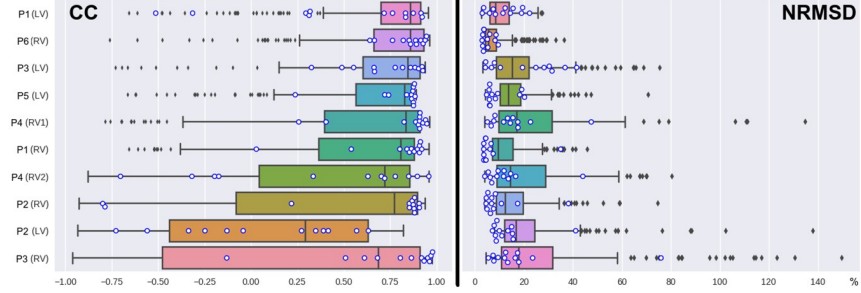

**Fig 1. The distribution of correlation coefficient (CC) and NRMSD values for the simulated and measured ECGs in BSM electrodes for each patient-specific model.** Patient codes (P1–P6) and paced ventricles (LV or RV) are shown in the left column. The boxes show median values and interquartile ranges. The whiskers show the minimal and maximal values without outliers (the 3-sigma rule). Outliers are shown as points outside the whiskers. Blue circles show values of the CC and NRMSD values for the 12 standard ECG leads.

cases (40%) had negative or near-zero values in the 25% percentile range for the CC. In contrast, NRMSD values had relatively low variability among BSM ECG leads.

The mean CC obtained for patients with non-ischemic cardiomyopathies was not significantly different from that of patients with structurally normal hearts (p < 0.39, the Mann-Whitney U-test). The difference in the correlations between patients with CRT devices and focal tachycardia was also statistically insignificant (p < 0.45, the Mann-Whitney U-test). We also observed a noticeable difference in the correlation between models with RV and LV pacing in the same patient in a few cases. The most striking example was the results for patient P2, from whom we obtained $r_{mean} = 0.77$ for RV pacing and $r_{mean} = 0.29$ for LV pacing. The NRMSD difference (12.56% for P2(RV) vs. 16.85% for P2(LV)) was not pronounced but still quite large. However, in general, CC differences between the RV and LV pacing models were statistically insignificant ($p < 0.16$, the Mann-Whitney U-test).

BSM electrodes with poor correlation ($r < 0.5$) between simulated and recorded signals were not randomly distributed over the surface of the human torso but showed well-structured patterns, as depicted in Fig 2. BSM electrodes with poor correlation ($r < 0.5$) tended to be grouped into an oval-shaped region on the left side of the torso (n = 6) and an elongated band-shaped region encircling the torso (n = 4).

The mean CC and NRMSD values for ECGs in 12-lead ECGs had somewhat greater variability among the models. The mean CC varied from 0.91 (case P4(RV1)) to 0.12 (case P2 (L2)), while the mean NRMSD varied from 4.47% (case P6(RV)) to 22.24% (case P3(RV)). Eight of the ten simulation cases (80%) had a mean CC $r_{mean} = 0.7$, and nine (90%) cases had mean NRMSDs lower than 15%. As with the BSM ECG, the CC varied significantly among the standard ECG leads (Table 4). The results of an accuracy assessment based on the 12-lead ECG were consistent with those obtained by BSM ECG in some cases. For example, the minimal mean of CC and NRMSD values for BSM and 12-lead ECGs were observed for the same cases P2(LV) and P6(RV), respectively.

However, a detailed analysis of the cases showed some differences. For example, correlation values for 12-lead ECGs were not always in the 25–75% percentile range for CCs of BSM leads (Fig 1). In particular, the SECG of the P2(RV) cases strongly correlated with the PECG in the 12-lead ($r_{mean} = 0.87$, $r_{25\%} = 0.69$), but signals from BSM leads were poorly correlated ($r_{mean} = 0.77$, $r_{25\%} = -0.08$). In contrast, the SECG for the P1(LV) case had a moderate correlation with the PECG in standard 12 leads ($r_{mean} = 0.74$, $r_{25\%} = 0.31$) but was well correlated in BSM leads

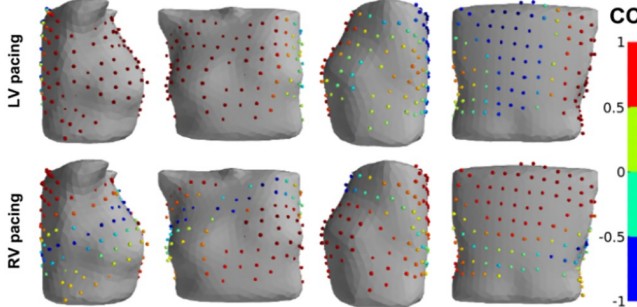

**Fig 2. An example of a body-surface electrode array (spheres on the electrode locations) with a color-coded map of the CC between simulated and measured ECGs for patient P1 at LV (upper panels) and RV (lower panels) pacing.** The color scale indicates a CC value from -1 (blue) for no correlation to +1 (red) for the highest correlation. The cases show two frequent patterns of regions with low correlation. The top panel (P1(LV) model) shows a single oval-shaped region of low CC on the left side of the spine, while the low panel (P1(RV) model) shows an elongated, band-shaped region of low CC around the torso.

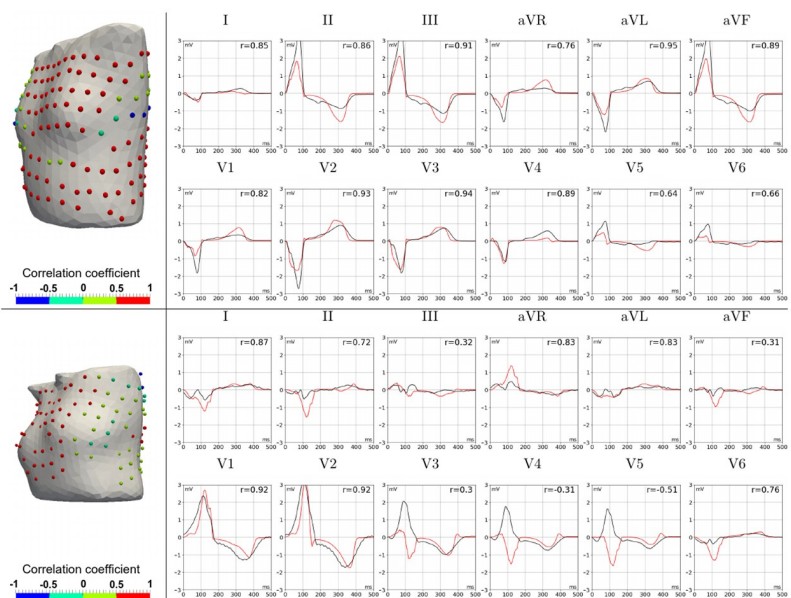

**Fig 3. Comparison of simulated and patient ECGs in P1(LV) (upper panels) and P5(RV) (lower panels) patient-specific models.** Color spheres in the left panels show CC values (see color-coding scale) for the BSM electrode array on the torso, and the right panels show simulated (red lines) and measured (black lines) ECG signals in standard 12-lead array. The top panel shows a good simulation of the P1(LV) patient data with high CC values (red spheres) in most BSM leads and good agreement between time-dependent signals in each standard lead. The bottom panel demonstrates the opposite QRS complexes in simulated and measured ECGs (see leads V3–V5) for the P5(RV) model as an example of possible simulation inaccuracy.

($r_{\mathrm{mean}} = 0.86$, $r_{25\%} = 0.70$). Certain cases (P5(LV), P1(RV), and P3(LV)) also showed tangible differences in NRMSDs between BSM and 12-lead ECGs.

Lead placement on the torso surface could explain the differences in correlation values, which were observed for BSM leads and standard 12-leads. For example, electrodes V3–V6 were in the area of poorly correlated BSM leads in case P1(LV), so the 12-lead ECGs underestimated the consistency of the SECGs and PECGs. In contrast, in case P2(RV), neither I, II, or III lead electrode positions nor lead electrodes V1–V6 were in the zones of poorly correlated BSM leads; in this case, the 12-lead ECG overestimated the similarity between the SECG and PECG.

Fig 3 presents a comparison of PECG and SECG in standard 12-leads. The most striking difference in ECG morphology was the opposite polarity of QRS complexes but correct T-wave polarity. This phenomenon was also observed in cases P1(LV) (leads V3–V6), P1(RV) (lead V1), P2(LV) (leads I and V1), and P2(RV) (leads I and V1). In a few cases, QRS complexes of the SECG were not the opposite of the PECG, but the QRS and T-wave magnitudes differed significantly. More pronounced differences (up to 3 mV for the QRS magnitude) were observed in cases P3(RV) (leads II, III, AVF, V3, and V4) and P4(RV1) (leads II, III, AVR, V4, and V5). In these cases, the QRS and T-magnitude were greater for simulated ECGs. However, this was not a general rule. In particular, in case P6(RV), the QRS and T-wave amplitudes of the SECG in leads II, AVF, V1, and V2 were lower than those in the PECG. In most cases, the QRS width and QT duration of simulated and real ECGs were well matched, but in a few cases, these values had substantial deviations in both directions.

In summarizing these results, we can conclude that the numerical ECG simulation using the conventional set of parameters provided relatively accurate results in most cases (80%). However, the accuracy of the simulation both in terms of correlation values and ECG

morphology in 12-lead ECGs had a significant level of variability, and variability was observed not only among different patients but also among various cardiac excitation patterns in the same patient.

## Feasibility of model parameter fitting

In this section, we analyze the effects of varying model parameter values on the accuracy of ECG simulation. We studied the feasibility of fine-tuning the model parameter values using single-parameter optimization. Variations in the epi/endo ratio, apicobasal heterogeneity, anisotropy ratio, and the lungs, blood, liver, and spine electrical conductivities were all tested. We selected the integrative RED between simulated and real ECGs throughout the BSM array for the signal comparison (see formula 15 for the distance $\Delta(x_{var})$ between the simulated and patient signals on the entire set $I$ of the BSM lead array). Fig 4 shows the dependencies of $\Delta(x_{var})$ upon each parameter $x_{var}$, where $\Delta(x_{var})$ was calculated at several $x_{var}$ values from the parameter range and interpolated using a cubic spline.

As is evident, the models are separated into two categories. First, four models in the first category (P1(LV), P1(RV), P3(LV), and P6(RV)) yield a RED value below 100%; the other six models in the second category yield a RED value over 100% for all parameter values within the study range. Models in the first group have relatively high CC between reference SECGs and PECGs. Despite the tangible effect of variation in parameter values, as well as their changes within the physiological range and study variation ranges they did not substantially increase the CC value.

Dependencies of the model output error (in terms of RED) on the parameter values had various patterns that differed significantly in the different modeling cases. The optimal values of parameters providing the local minimum error of the model output within the study range were found in only a few cases. The local minimum of the RED function within the study range existed in eight cases (80%) for the endo-epi ratio, in five cases (50%) for apicobasal heterogeneity, and in three cases (30%) for the anisotropy ratio and lung conductivity. There was no local minimum of the RED in any model with varying liver, blood, and spine conductivity.

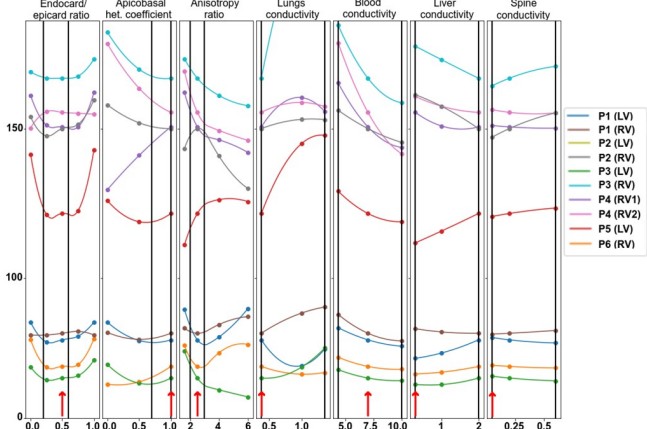

**Fig 4. Dependence of the integrative distance (RED) between simulated and patient ECGs on each varied model parameter in the patient-specific models.** Circle marks show the RED for the computed models at certain parameter values, and dependencies within the range intervals are interpolated. Reference parameters are annotated with red arrows near the bottom of each plot. Vertical lines indicate the physiological range of parameter variation. The 0% level indicates no difference between the model and simulated data. Color annotation indicates the patient cases (P1–P6) and the position of the electrode stimulation (LV or RV). Model P2(LV) shows outlier behavior and is excluded from the visualization.

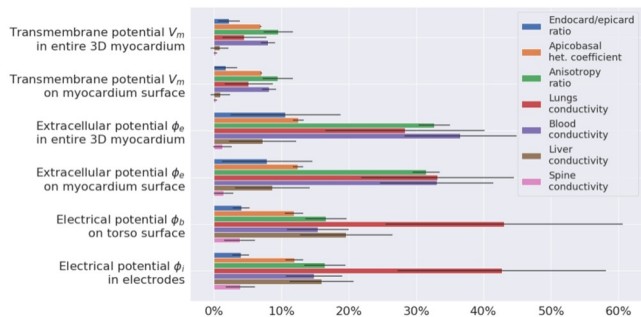

**Fig 5. Integrative effect of parameter variation in the physiological range on the model output signals (see annotation in the left column).** Different colors indicate individual parameters under variation (see legend). The effect was assessed using RED metrics with respect to output from the reference models. Bars show mean values across ten patient models, and error whiskers show standard deviations.

For these conductivity, the RED reached the minimal and maximal values at the borders of the study range. We also noticed that optimal parameter values existed in the model cases in the first category of models, which generally showed less error in the model output.

Setting the optimal parameter values (taken within or at the boundary of the study interval instead of the reference one) did not significantly improve the accuracy of the simulation results. One-dimensional optimization of the anisotropy ratio reduced the error by 4–7%, lung conductivity by 2–10%, and the blood conductivity by 3–5%. Optimizing other parameters reduced the error by less than 1%. Therefore, the results suggest that one-dimensional minimization does not allow fine-tuning of model parameters to real ECGs.

**Integrative effects of parameter variation on electrical potentials.** In this section, we estimate the average effects of a substantial variety of model parameters on different model signals in all the patient case models (a population of models). We evaluated parameter variation effects on the following model signals: transmembrane potentials ($V_m$) in the myocardium of the entire ventricular wall ($I = \{i|i \in \text{grid\_points}(\Omega)\}$) and on the ventricular surface ($I = \{i|i \in \text{grid\_points}(\partial\Omega)\}$), extracellular potentials ($\phi_e$) in the myocardium volume ($I = \{i|i \in \text{grid\_points}(\Omega)\}$) and on the myocardial surface ($I = \{i|i \in \text{grid\_points}(\partial\Omega)\}$), potential ($\phi_b$) on the torso surfaces ($I = \{i|i \in \text{grid\_points}(\partial\Omega_b)\}$), and ECGs computed in the BSM electrode array. For each varied parameter $x_{\text{var}}$, we computed $\Delta_{x_{\text{var}}}$ using the respective reference model signals (**R**) in formula (16) to calculate the maximal RED within the physiological range of parameter variation for the tested model outputs in each case model. Fig 5 shows the results of the analysis in terms of means and standard deviations in the model population.

In general, the intensity of the response to parameter variation was different between the case models, varied parameters, and model outputs. However, a few regular features were observed. For every parameter variation, the relative effects on transmembrane potentials on the ventricular surface were almost the same as those produced throughout the myocardium. Thus, we showed that surface-mapping parameter sensitivity is representative of the entire tissue. Similarly, relative parameter effects on extracellular potentials on the myocardial surface were similar to those throughout the myocardium, and effects observed in the BSM electrode vest were similar to those on the entire torso. Thus, the BSM electrode vest with 224 electrodes can be considered representative of the electrophysiological activity on the entire torso surface. The transmembrane potential showed the lowest sensitivity to variations of the model parameters. Only three of the seven parameters we tested induced changes of over 5%: apicobasal heterogeneity, anisotropy, and blood conductivity. Responses of transmembrane potential to their variation did not exceed 10%. The effects of parameter variation on myocardial

extracellular potentials were higher than those on the transmembrane potentials
($p = 0.03 < 0.05$ for all varied parameters, the Mann-Whitney U-test). The following parameters showed the strongest effects on myocardial extracellular potentials: the anisotropy ratio ($33\pm2\%$), lungs ($28\pm12\%$), and blood conductivity ($37\pm8\%$). The effects of the other parameters were less than 15%.

The effects of parameter variation on the torso surface potentials can be classified into three groups. Lung conductivity demonstrated the highest effect ($43\pm18\%$). Variations in apicobasal heterogeneity, the anisotropy ratio, blood conductivity, and liver conductivity all produced a medium effect ($12\pm2\%$, $16\pm3\%$, $15\pm5\%$, $20\pm7\%$). Variations in transmural heterogeneity and spine conductivity had effects of less than 7%. The effects of liver conductivity on extracellular potentials were higher on the torso surface than on the ventricular surface ($19\%\pm6\%>8\%\pm5\%$, p<0.0014, Mann-Whitney test). The highest variability in $\Delta_{x_{var}}$ among the patient case models was from lung conductivity variation. The standard deviation ranged from 3.6% for the effect on transmembrane potentials to 17.6% on torso potentials. For any other parameter variation, variability in $\Delta_{x_{var}}$ between the models was less than 10%, with an essential standard deviation of 8.29% for the effect of blood conductivity variation on myocardial extracellular potentials and a standard deviation of 6.89% for the effect of liver conductivity variation on torso potentials.

## Effects of parameter variation on properties of myocardial depolarization and repolarization

In this section, we analyze the sensitivity of several characteristics of ventricular repolarization and depolarization to model parameter variation. We studied several model properties: late activation time (ms), the dispersion of APD (ms), the dispersion of repolarization time (ms), the maximal extracellular potential amplitude on the myocardium surface during depolarization (mV), the maximal extracellular potential amplitude on the myocardial surface during repolarization (mV), the peak of the QRS complex (maximal potential during depolarization) on the torso surface (mV), and the peak of the T-wave (amplitude of potential during repolarization) on the torso surface (mV). The effect of individual parameter variation in each case model was assessed as a min-max diapason of the biomarker within the physiological range of the parameter relative to the reference value produced by the reference model (see formulas (17)–(18)). Table 5 shows the reference values of all ten case models. Fig 6 shows the relative diapasons for the action potential properties, and Fig 7 shows the extracellular potential properties.

**Table 5. Characteristics of ventricular repolarization and depolarization in the reference models.**

|  | P1 (LV) | P1 (RV) | P2 (LV) | P2 (RV) | P3 (RV) | P3 (LV) | P4 (RV1) | P4 (RV2) | P5 (LV) | P6 (RV) |
|---|---|---|---|---|---|---|---|---|---|---|
| Late activation time (ms) | 164 | 132 | 137 | 136 | 114 | 170 | 152 | 149 | 108 | 105 |
| APD dispersion (ms) | 57 | 55 | 52 | 52 | 49 | 53 | 51 | 48 | 55 | 48 |
| Repolarization dispersion (ms) | 141 | 99 | 116 | 98 | 80 | 153 | 145 | 123 | 62 | 107 |
| Max. amplitude of extracellular potential in depolarization (mV) | 33.7 | 29.4 | 32.1 | 29.1 | 31.7 | 32.4 | 38.3 | 33.9 | 27.9 | 27.5 |
| Max. amplitude of extracellular potential in repolarization (mV) | 17.4 | 9.7 | 13.7 | 8.9 | 9 | 14.3 | 16.3 | 11.9 | 4.9 | 7.8 |
| Max. absolute amplitude (mV) | 3.7 | 4.4 | 3.2 | 3.2 | 3.4 | 3.3 | 2.9 | 4 | 2.5 | 2 |
| Max. absolute T-wave amplitude (mV) | 1.9 | 2.2 | 1.7 | 1.6 | 1.8 | 2 | 2.4 | 2 | 0.7 | 1.7 |

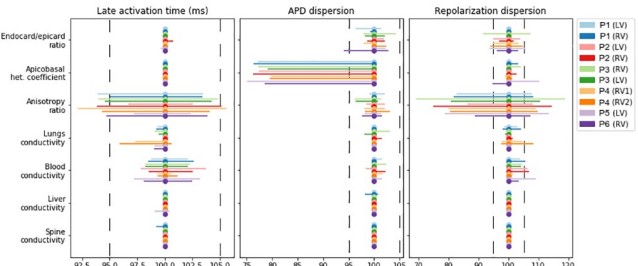

**Fig 6. Variation in the temporal characteristics of depolarization and repolarization (see annotation on top of the panels) in the models under univariable parameter variation (see annotation in the left column); 100% on the horizontal axis indicates the reference model values.** The bars show the spans of characteristics from the minimum to maximum relative to the reference. Color annotation indicates the patients (P1–P6) and the position of the electrode stimulation (LV or RV). Vertical dashed lines show a ±5% deviation from the reference.

Variation in the anisotropy ratio substantially affects the late activation time (92.1–105.6%) and dispersion of repolarization (70–119%). Variation in the latter was quantitatively higher in most of the models. As expected, variation in the apicobasal heterogeneity coefficient affected APD dispersion (75–100%) and the T-wave amplitude (81%–143%). However, it had a minimal effect on the repolarization dispersion or amplitude characteristics of extracellular potentials in all models. Similar to the integrative effects of model parameter variation on the overall extracellular potential signals, the most pronounced effects on the characteristics of the myocardial extracellular potential and body-surface ECG were produced by the anisotropy ratio, lung conductivity, and blood conductivity. Variation in lung conductivity produced the most pronounced effects on the maximal extracellular potential on the myocardial surface during depolarization and repolarization, while variation in blood conductivity affected the amplitudes of QRS and T-wave complexes in the ECG on the torso surface.

**Regional sensitivity of extracellular potential on heart surface to parameter variation.**
In the two previous sections, we analyzed overall sensitivity to parameter variation of reference signals throughout the myocardial volume or surface. We found that the local sensitivity of the extracellular potential on the heart surface varied substantially in different regions of the ventricles. In this section, we focus on the regional features of the extracellular potential response to parameter variation in our patient-specific models.

For each varied parameter, we compared extracellular potential signals from the reference model and the model with optimal parameter, which provided the maximal overall RED value for the entire myocardial surface (see formula (16), $I = \{i | i \in \text{grid\_points}(\partial\Omega)\}$). Then, we built a regional RED map (sensitivity map) between the local signals for every point on the surface

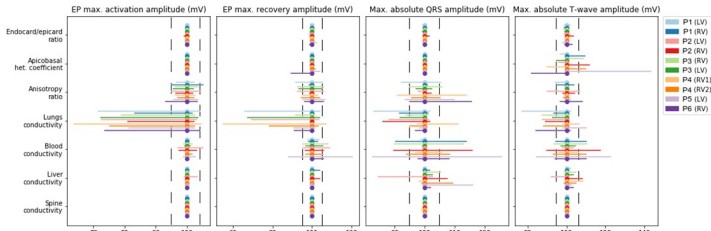

**Fig 7. Variation in the amplitude characteristics of depolarization and repolarization (see annotation on top of the panels) due to univariable parameter variations (see annotation in the left column) in the models.** The figure design is the same as in Fig 6.

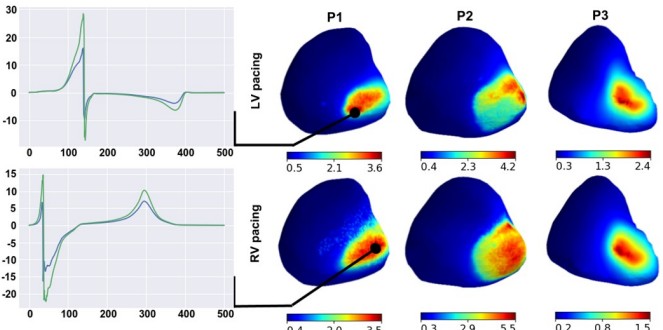

**Fig 8. Examples of regional RED maps for extracellular potential signals under variation in the liver conductance.** Maps are shown in the right panels for patient models P1–P3 with LV (upper panels) or RV (lower panels) pacing. A color map of every point on the ventricular surface shows the local RED scaling between the minimal and maximal values on the surface (see color scales on the bottom of the maps). Plots on the left show ECG signals from the reference model (green lines) and the model of maximal overall RED (blue lines). The signals are shown for point of the highest local RED on the map.

of the heart, thus scaling the effects of parameter variation between the ventricular regions. In Figs 8–11, we show representative examples of such regional RED maps for extracellular potential signals on the heart surface while varying certain parameters in patient case models. The cases are arbitrary, and if not specifically described, the map patterns for the rest of the models have similar features.

The effect of variation in liver conductivity on the heart surface extracellular potential has a well-structured RED map with small compact zones of high REDs (Fig 8, right panels). The regions of strong-to-moderate parameter influence are at the basal segments of the epicardial posterior wall of the right ventricle for all ten patient case models, not capturing the endocardial surface of the ventricles. In these regions, parameter variation affects the extracellular potential peaks, but the signal polarity and peak timing do not change (Fig 8, left panels).

The regional RED map of the effect of apicobasal heterogeneity variation also has a well-structured pattern (Fig 9, right panels). Two regions of high REDs are localized on the epicardial surface. In all ten cases, the first region is close to the apex, and the second is close to the point of the initial activation. In the region of the activation point, variation in apicobasal heterogeneity shifts the time to a T-wave peak on unipolar electrocardiograms but does not affect the maximal amplitude (Fig 9, left panels, upper frames). On the heart apex, the effects are

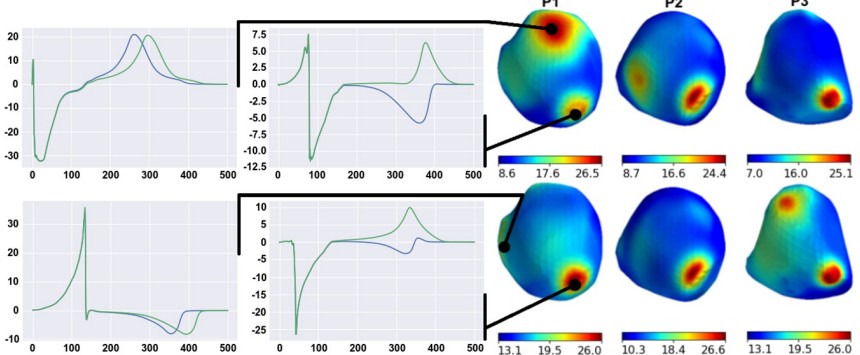

**Fig 9. Examples of regional RED maps for the extracellular potential under variation of the apicobasal heterogeneity coefficient.** The figure design is the same as in Fig 8.

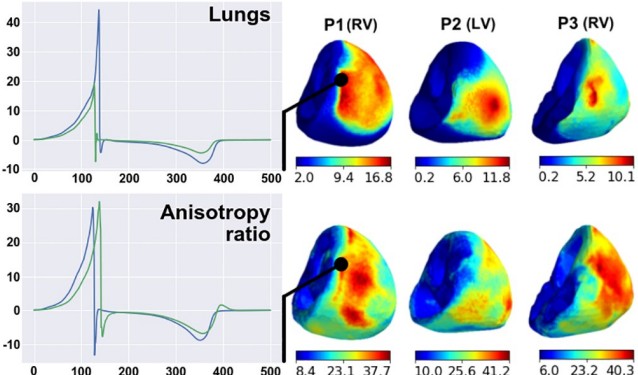

**Fig 10. Examples of regional RED maps for the extracellular potential under variation in lung conductivity (upper panels) and the myocardial anisotropy ratio (lower panes) for the P1(RV), P2(LV), and P3(RV) models.** The figure design is the same as Figs 8 and 9.

more substantial. Parameter variation causes inversion in the T-wave and a shift in time to the T-wave peak (Fig 9, left panels, lower frames). For every case model, the parameter does not affect the ECG QRS complex.

The regional effects of other parameters on the ventricular extracellular potential are not spatially compact, as described above (Figs 10 and 11). Regions with a high effect of lung conductivity variation and anisotropy ratio variation are on the epicardial surface close to the lungs in all ten cases (Fig 10, right panels) and on the endocardial surface in five of the ten cases. Simultaneously, lung conductivity variation significantly affects the peaks of the extracellular potential, but the anisotropy ratio does not (Fig 10, left panels). Regions with high REDs on the regional maps for the endo/epi ratio and blood conductivity variation are co-localized in the late activation zone of ventricles in seven of the ten cases (Fig 11), except P4 (RV1), P4(RV2), and P5(LV).

**Regional sensitivity of extracellular potential on torso surface to parameter variation.** We used the same approach as the previous section to build regional RED maps for potentials on the torso surface ($I = \{i | i \in \text{grid\_points}(\partial\Omega_b)\}$). The results are shown in Figs 12 and 13. Unlike parameter effects on the heart surface extracellular potential, the regional pattern of effects on the torso significantly varied between the patient models.

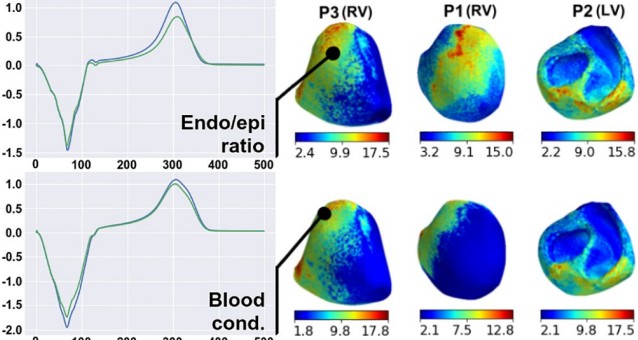

**Fig 11. Examples of regional RED maps for the extracellular potential under variation in the epi-endo heterogeneity coefficient (upper panels) and blood conductivity (lower panels).** The figure design is the same as in Fig 10.

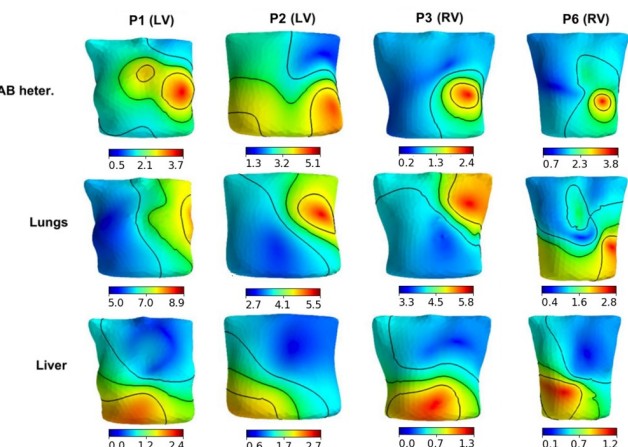

**Fig 12. Examples of regional RED maps for the torso surface potential under variation in the apicobasal heterogeneity coefficient (upper panels), lungs conductivity (middle panels), and liver conductivity (lower panels) in four patient-specific models (see annotation on the top).** The design of the map visualization is the same as in Figs 8–11.

Variations in the apicobasal heterogeneity coefficient and lung conductivity showed similar regional RED patterns, with a substantial effect on the ECG on the left side of the torso in all ten cases for the former parameter and seven of the ten cases for the latter (Fig 12, top and middle rows). These regions include the locations of standard chest leads. The liver conductivity variation showed a notable effect on either the left side of the torso or on the anterior region of the torso (Fig 12, bottom row). Regional RED maps under anisotropy ratio and blood conductivity variation are similar to those shown for variation in the apicobasal heterogeneity coefficient in seven of the ten cases, while three models show specific features of the RED for each parameter variation (not shown). Spine conductivity variation produced a small effect and an almost-uniform regional RED map (not shown). Variation in the transmural heterogeneity coefficient showed various spatial RED patterns for the torso potential for different patient-specific models (Fig 13). While these patterns differed across models, they were similarly independent of the paced ventricle (LV or RV) for each model (compare upper and lower rows in Fig 13).

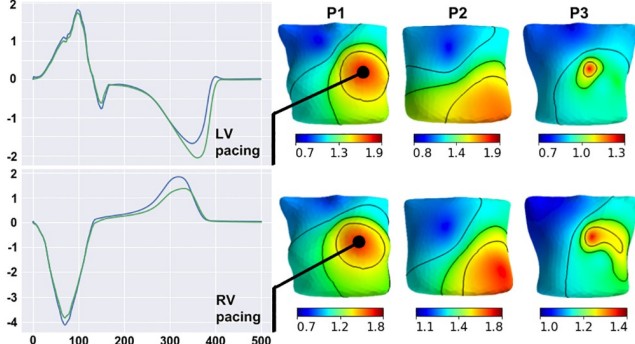

**Fig 13. Examples of regional RED maps for the torso surface potential under variation in the transmural heterogeneity coefficient in three patient-specific models (see annotation on the top).** The top row shows the pattern for LV pacing, and the bottom row shows the pattern for the RV pacing. The design of the map visualization is the same as in Figs 8–11.

## Discussion

In this study, we attempted to test the accuracy of state-of-the-art methods of personalized ECG simulation with standard 12-leads and an electrode array on a torso surface. For this purpose, we applied simulation tools, including the bidomain model, the common TNNP model of human ventricular cardiomyocytes accounting for heterogeneity in the cellular properties, a rule-based approach to model myocardial fiber orientation, and a personalized CT-based geometry of the heart, torso, and internal organs with different electrical conductivities. In this study, the His-Purkinje system was excluded from the simulation. Instead, we simulated the focal type of ventricular excitation using precisely determined patient-specific positions of excitation origins in patients with implanted CRT devices and focal ventricular tachycardia.

When choosing model parameters to vary in the sensitivity analysis (see Table 3), we focused on those parameters whose effects were not analyzed in detail in previous studies. We did not analyze the effects of the ionic parameters on simulated ECGs as model sensitivity to their variation and effects on ECG characteristics were previously studied in several works in detail [23]. However, coefficients of epi/endo and apicobasal heterogeneity were included in the analysis because of their essential role in the T-wave orientation was previously demonstrated in the model at the normal activation sequence, while no data were available on their significance for ECG morphology in cases of ectopic action. It is known that the ratio of cell membrane capacity to the surface-to-volume ratio affects strongly conduction velocity. In our work, we have tried to maintain a realistic conduction velocity for all simulation cases, so we did not vary the membrane capacitance and the surface-to-volume ratio. When varying the parameters of anisotropic electrical conductivity of the myocardium, we used a special Roth et. al. mathematical framework that establishes a relationship between extracellular and intracellular conductivities. This approach allows us to vary the values of four electrical conductivity parameters based on varying the value of only one parameter, the anisotropy ratio ($\lambda_L/\lambda_T$). We used a variation of this parameter instead of variations of the four parameters for our sensitivity analysis. We also varied the conductivity of the internal organs and blood, intending to study the regional effects of these variations.

### Accuracy of ECG simulation

The results of the ECG simulation accuracy analysis were ambiguous. The simulation provided a high mean correlation between the patient and simulated ECGs of over 0.7 for most models (80%), which is conventionally interpreted as a strong correlation [32], and a relative error (NRMSD) of less than 15% in 60% of the models (see Fig 1, Table 4). The correlation level was similar to that reported in [9, 11] and slightly lower than the data reported in [20]. However, the accuracy was highly variable among simulation cases, the mean correlation values and mean NRMSD varied from $r = 0.86$ and 5.6% in the best case, respectively, to $r = 0.29$ and 18.1% in the worst case, respectively (Fig 1). Moreover, in some cases, the accuracy metric values varied significantly among ECG leads. In particular, there were near-zero and even negative correlations in some BSM array leads. Besides these quantitative differences, a qualitative difference in the morphology of the simulated and patient ECGs was observed. For example, the opposite polarity of QRS complexes of the simulated ECG in one or more standard leads was detected in four of the ten cases (Fig 3).

Therefore, our results show that excluding uncertainties in the model related to His-Purkinje conduction does not improve accuracy in a simulated personalized ECG using a population-based set of model parameters. The results suggest that carefully tailoring model parameters is necessary to use the models in clinical applications.

In this study, ECG simulation was performed for a heterogeneous group of patients, which included patients with non-ischemic cardiomyopathies and normal myocardia (Table 2). We cannot ignore that the model assumptions did not account for specific myocardial remodeling in patients with cardiomyopathies, so we expected to obtain less simulation accuracy in cardiomyopathy patients. However, we found no statistically significant differences in accuracy between patients with cardiomyopathy and those with structurally normal hearts.

One of our most interesting observations was varying degrees of accuracy in the SECG for right and left ventricular pacing in the same patients. This observation indicates that simulation accuracy depends on the pattern of excitation of the heart. The effect of the excitation pattern was also patient-specific, and there was no significant difference in the average correlation between the right ventricular and the left ventricular pacing subgroups.

The difference between PECGs and SECGs with standard 12-leads varied between the leads and were patient-specific. This was observed for QRS width, QT duration, and other clinically significant properties of ECG signals. Such irregular simulation errors do not support any assumptions about certain systematic errors in the model or missing a few important model parameters. Most likely, the reason for the personal ECG simulation discrepancy lies in the use of population-based parameters, which must be individually adjusted to reproduce specific clinical data.

What level of agreement between model output and data is needed for the model to be useful in clinics? Theoretically, modeling errors should not exceed the typical ECG recording error (e.g., electrical and muscle noise, changes in the contact resistance of the ECG electrode with the skin, inaccuracies in determining the positions of standard ECG leads (V1-V6)) and the level of ECG variability caused, for example, by the patient's breathing. However, each medical application requires its own level of modeling accuracy, which must be identified through clinical research. Nevertheless, we may assume that 4 out of 10 simulation cases provided accuracy close to that required for clinical applications. This shows that the required level of model accuracy is fundamentally achievable. However, modeling using averaged population-based parameters does not work well, and this approach cannot be directly translated into clinical practice.

## Univariable optimization of model

Theoretically, if real ECGs were available, optimal individual values of model parameters would provide the minimum difference between real and simulated ECGs. This requires solving the problem of multiparametric optimization. In our study, we chose seven parameters of the cardiac electrical activity model to analyze the model's sensitivity to their variation. The first three parameters were related to myocardial properties, and the other four parameters were the electrical conductivities of blood and internal organs. For this purpose, we built one-parametric function for the error between SECG and PECG for each optimization parameter independently of the others and determined whether the solution of a one-dimensional minimization problem improved simulation accuracy (Fig 4).

Numerical experiments revealed that the univariable fit in the physiologically plausible interval of parameter values did not significantly improve the accuracy of model personalization. In particular, the variations in the parameters did not affect the qualitative classification of the models into groups with 'high' and 'low' accuracy in terms of CC metrics. Moreover, the local minima of one-parameter dependencies were mostly not located within the permissible parameter intervals (Fig 4).

Several explanations exist for this fact. Models with pathological remodeling may require a wider range of parameter variations. However, our models may not have considered factors that significantly affect simulation accuracy, such as myocardial fibrosis and epicardial fat. We

also cannot exclude anatomical errors of the models associated with shifts in the positioning of the heart and internal organs during the CT procedure and ECG recording, such as movement due to respiratory movements of the chest. The simulations also used the geometry of heart ventricles during diastole, but heart contractions were not considered. Finally, the possibility of retrograde activation of the Purkinje fiber system during pacing, which could alter myocardial activation patterns, cannot be eliminated.

The third and most likely reason for the low efficiency of single-parameter optimization was the nonlinear behavior of model output upon the parameter values. Consequently, the objective function was non-convex, as indicated by the absence of local minima for most of the one-dimensional problems. If the objective function is non-convex, only multiparametric optimization based on special algorithms can fine-tune the parameters.

### Sensitivity of model output to variation in model parameters

Ranking model parameters according to their impact on model outputs can be useful for multiparametric optimization. Model parameters with a weak effect on the model outputs may be excluded from the fitting process. This approach could reduce the dimensionality of the optimization problem. In this study, we used a one-at-a-time approach for the model sensitivity analysis in terms of maximal deviation from the reference signals. Reviews [33, 34] have criticized such a simple approach and recommended the more advanced approach of a global sensitivity analysis based on exploring the multidimensional parameter space. However, this would require significantly more computational power, especially for the bidomain model of the myocardium; nevertheless, the lack of a global sensitivity analysis is a limitation of this study and a task for further investigation.

The results of the one-at-a-time sensitivity analysis, in addition to the analysis of the integrative distance between the simulated ECG and experimental data, showed that myocardial anisotropy, the apicobasal heterogeneity coefficient, lungs, and blood conductivity had the greatest effects on model outputs (Fig 5). The transmembrane potential was most strongly influenced by variations in myocardial anisotropy, apicobasal heterogeneity, and blood conductivity, while the electrical conductivity of blood and lung tissue most strongly affected the cardiac extracellular potential. In addition, variation in lung conductivity exerted a greater influence than all the other model parameters on the body-surface potential. These data are consistent with the results of other studies [9, 10].

In addition to the effects of parameter variation on overall time-dependent signals, we evaluated specific effects on the physiologically significant characteristics of the signals (Figs 6 and 7, Table 5). Generally, variations in model parameters have three main ways of influencing cardiac electrical activity: change in the velocity and geometry of the excitation wavefront, change in the APD, and change in the amplitude of extracellular potentials. We showed that parameter effects on the temporal characteristics and amplitude of action potential and extracellular potentials were different between the parameters and signal biomarkers, enabling a classification of model outputs in terms of sensitivity to tested model parameters.

Myocardial anisotropy had the most significant effect on conduction velocity, giving it a strong influence over the latest activation time and repolarization dispersion (Fig 6). An increase in myocardial anisotropy increases the late activation time for the focal stimulation of the myocardium. These effects are consistent with other studies on idealized models of the left ventricle [25]. As expected, increasing apicobasal and transmural myocardial heterogeneities increases APD and repolarization dispersion (Fig 6).

As shown in previous research [9], apicobasal cellular heterogeneity is necessary for the correct orientation of the ECG T-wave under normal ventricular activation from the His-Purkinje

system. In that work, a shorter action potential on the ventricle apex and a longer action potential allowed the model to reproduce a T-wave concordant with the QRS complex. In contrast, we simulated the activation of ventricles from a focal source. In this case, the directions of the QRS wave and T-wave were always opposite in both the clinical data and nearly all the simulations (Fig 3). With the point activation considered in this study, the direction of the repolarization wave coincides with the direction of the depolarization wave, and the apicobasal heterogeneity coefficient does not affect the ECG T-wave orientation but does influence the T-wave amplitude.

Models with heterogeneous cellular properties in the ventricular walls include endocardial and epicardial layers occupying the wall depth at a ratio anywhere from 30%:70% to 60%:40%. Variation of the endo-epi ratio in the range slightly affected SECG accuracy (Fig 4), the homogeneous models with only epi- or endo-type cells revealed a significant difference between the PECG and SECGs.

Electrical conductivity of the internal organs and blood affected the amplitude of extracellular and torso surface electrical potentials (Fig 7). The results we obtained for focal myocardial activation are consistent with results from other simulation studies [35, 36], showing the effects of the conductivity of the medium in which the myocardium is placed on the conduction velocity of the excitation wave.

Unlike the monodomain model, the bidomain model with bath considers the influence of the interface conditions on the heart and torso surface potentials (equal currents through the border between the myocardium and the torso $\sigma_{el}\frac{\partial \phi_e}{\partial \mathbf{n}} = \sigma_b \frac{\partial \phi_b}{\partial \mathbf{n}}$ on $\partial\Omega$). A method of segmentation of the myocardium and internal organs thereby played a significant role in the bidomain model simulation. Some studies have utilized segmentations without a gap between the myocardium and organs [9, 23, 24], while others have suggested a gap of 0.3–2.0 cm [10, 11, 21, 37]. In the latter case, the conductivity of elements in the gap is equivalent to the generalized conductivity of the torso, while the former approach to segmentation is more realistic because the left lung closely adjoins a certain region of the left ventricular epicardium and is separated from it by two pericardial sheets with a thin 0.5–1 mm layer of pericardial exuded. The second approach is more convenient since it simplifies the application of the boundary element method and allows easy mixing or manual correction of voxel models of adjacent organs simultaneously. However, the second approach assumes the presence of a zone with generalized torso conductivity between the myocardium and other organs, which may lead to inconsistencies in simulation results, so we used the first method of heart-torso segmentation. The cardiac surface contacts the lungs, blood, and liver, so we expected that changes in the conductivity of these organs would affect the conduction velocity and dispersion of repolarization in the myocardium. However, according to our results, variations in the physiologically acceptable range of liver conductivity in every case model and lung conductivity in most of the models had negligible effects on late activation time and repolarization dispersion. Only variation in the electrical conductivity of blood had a notable effect on late activation time and dispersion of repolarization (about ±5%, (7)).

However, variations in internal organ conductivity had significant effects on amplitudes of the extracellular potential on the myocardium surface and the body-surface potential. In particular, an increase in lung conductivity over the reference value led to decreases in the QRS complex and T-wave amplitude in every patient model.

## Regional sensitivity of model output to variation in model parameters

We found that variation in certain model parameters led to specific patterns in regional sensitivity maps on the myocardium and torso surfaces (Figs 8–13). Variation in lung conductivity

manifested predominantly on the epicardium of the anterior or anterior-septal region of the ventricles and on the left lateral regions of the torso (Figs 10 and 12). The effect was stronger on the left side of the torso because the left lung contacts a larger part of the ventricle surface than the right lung.

A specific regional pattern in the RED map was also observed for variation in apicobasal heterogeneity in the form of two zones in the apical and basal regions of the ventricles (Fig 9). This pattern reflects the APD gradient along the longitudinal ventricular axis from the apex to the base, which results from the slope of the linear model of the apicobasal heterogeneity in the $I_{Ks}$ current we used. The highest effect of liver conductivity variation was in the heart and torso regions near the heart (Figs 8 and 12). In contrast to the focal regional effects of the liver con-ductivity and apicobasal ratio variation, the effect of lung conductivity was not so compact, as the lungs occupy a significant volume of the chest and their surface surrounds most of the heart.

### Possible reasons for model shortcomings

In this study, we showed that using population-based parameters of the bidomain model with-out personal tailoring was the principal factor causing the inaccuracy in our modeling results. However, we should at least briefly consider other possible reasons for model shortcomings, such as the inaccurate detection of the myocardial activation point, retrograde activation of the ventricular conduction system, geometrical discrepancies related to segmentation and meshing of the internal organs, and not accounting for some details in the model's anatomical structure.

In short, we determined the origins of focal tachycardia with a method that was less accu-rate than simply detecting the position of the tip of the stimulating electrode by CT. However, we found no specific differences in simulation accuracy in these two groups of patient models. This observation does not support the significance of precisely localizing the early activation zone for model outputs. The hypothesis on the essential contribution of retrograde activation of the His-Purkinje system at focal ventricular activation seems valuable and needs to be assessed in future studies.

Inaccuracy in organ segmentation makes a rather small contribution to the modeling results because minor changes in organ borders do not lead to significant changes in their vol-umes or percentage of the whole torso volume. Errors in heart segmentation may affect model output much more because the ECG amplitude is almost linearly dependent on the mass of the myocardium [38]. However, not accounting for changes in ventricular geometry during the cardiac cycle may exacerbate segmentation errors [24] and have a greater impact on the simulation results. Mesh refinement also affects the activation time and conduction velocity of the excitation wave [39]. We used the Oxford Chaste solver, which shows the low dependency of the solution on mesh element size [39]. We also improved mesh quality using refining-by-splitting to a number of elements where further refinement would not have led to a change in the solution at any point by more than 5%.

In summary, the most significant sources of simulation discrepancy with clinical data are the choice of model parameters and not accounting for structural features of the pathological myocardium, which should be tailored to personal data.

### Limitations

Our study has several limitations. First, it does not consider anatomical structures that may affect the accuracy of the cardiac electrical field, particularly epicardial fat, fibrosis, the ster-num, and the ribs. We used the TNNP 2006 model to simulate action potential in human

cardiomyocytes, which has a steeper repolarization profile than in reality; other human ventricular action potential models (e.g., [40]) can be utilized as well. Moreover, the cellular models we used in this study did not account for possible remodeling in the cellular mechanisms of excitation, which may affect action potential profiles in patients with myocardial pathology. We also assessed the effects of varying only seven model parameters; this list can be extended to account for the rather high inconsistency of values reported for several of our parameters. We adopted a one-at-a-time approach to sensitivity analysis, so global methods of sensitivity analysis [33, 34] can be further applied to explore the multi-parameter space based on our findings. Finally, ventricular contraction can reshape tissue geometry and may contribute to electrical activity of the myocardium and ECG morphology.

## Conclusions

In this study, we evaluated the accuracy of cardiac electrical activity simulation in patients with implanted CRT devices and focal ventricular tachycardia, and we investigated model output sensitivity to variation in seven model parameters. Despite the relatively high average correlation between simulated and real ECGs, which was found in the numerical tests, certain simulations had significant errors. Models with averaged, population-based parameter values do not allow accurate personal ECG modeling without specific multiparametric tailoring. We also found that model tailoring to reproduce 12-lead ECGs may be less accurate than basing it on body-surface-array ECGs. Moreover, the level of model discrepancy depends on ventricular excitation timing, so various stimulation patterns should be considered in model tailoring where possible.

We found that variations in the myocardial anisotropy ratio, blood conductivity, and apico-basal heterogeneity had the strongest influences on the transmembrane potential, while variation in lung conductivity had a maximal influence on the body-surface ECG. The anisotropy ratio predominantly affected the latest activation time and the repolarization time dispersion. Apicobasal heterogeneity mainly affected the dispersion of APD. Variation in lung conductivity mainly changed the amplitude of the ECG. Fine-tuning model parameters using patient ECGs must be performed using multiparametric optimization with algorithms designed for non-convex optimization. If local cardiac electrograms and cardiac electrical activation maps are available, analyzing their temporal and amplitude characteristics and the spatial patterns of their discrepancies with simulations can help tailor the model parameters.

## Supporting information

**S1 File.**
(PDF)

## Author Contributions

**Conceptualization:** Vitaly Kalinin.

**Data curation:** Sukaynat Gitinova, Oleg Sopov.

**Funding acquisition:** Konstantin Ushenin, Olga Solovyova.

**Investigation:** Konstantin Ushenin, Sukaynat Gitinova, Oleg Sopov.

**Methodology:** Vitaly Kalinin, Olga Solovyova.

**Project administration:** Vitaly Kalinin.

**Software:** Konstantin Ushenin.

**Supervision:** Vitaly Kalinin, Olga Solovyova.

**Visualization:** Konstantin Ushenin.

**Writing – original draft:** Konstantin Ushenin.

**Writing – review & editing:** Vitaly Kalinin, Olga Solovyova.

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
