## [Decision Letter · Decision Letter 0]

28 Apr 2020

PONE-D-20-04786

Parameter variation in personalized electrophysiological models of human heart ventricles

PLOS ONE

Dear Mr. Ushenin,

Thank you for submitting your manuscript to PLOS ONE. After careful consideration, we feel that it has merit but does not fully meet PLOS ONE’s publication criteria as it currently stands. Therefore, we invite you to submit a revised version of the manuscript that addresses the points raised during the review process.

Please address comments indicated by the Reviewers.

We would appreciate receiving your revised manuscript by Jun 12 2020 11:59PM. To enhance the reproducibility of your results, we recommend that if applicable you deposit your laboratory protocols in protocols.io, where a protocol can be assigned its own identifier (DOI) such that it can be cited independently in the future. For instructions see: http://journals.plos.org/plosone/s/submission-guidelines#loc-laboratory-protocols

We look forward to receiving your revised manuscript.

Kind regards,

Elena G. Tolkacheva, PhD

Academic Editor

PLOS ONE

Journal Requirements:

"The development of personalized computer models was performed as part of the

project that is supported by the Russian Foundation for Basic Research (No. 18-31-

00401; KU; https://www.rfbr.ru/rffi/ru/ ). Computational resources, clinical data

preprocessing, software development were covered by the following research

programs (Institute of Immunology and Physiology Ural Branch of Russian Academy of

Sciences; theme No AAAA-A18-118020590031-8; OS; http://www.uran.ru/ ), (Ural

Federal University; RF Government Act \\#211 of March 16, 2013 (agreement

02.A03.21.0006); OS; https://urfu.ru/en/ ), (Institute of Immunology and Physiology

Ural; Branch of Russian Academy of Sciences; OS; http://www.uran.ru/). The funders

had no role in study design, data collection and analysis, decision to publish, or

preparation of the manuscript." 

We note that one or more of the authors are employed by a commercial company:"EP Solution SA,!

Reviewers' comments:

Reviewer's Responses to Questions

**Comments to the Author**

1. Is the manuscript technically sound, and do the data support the conclusions?

Reviewer #1: Partly

Reviewer #2: Yes

Reviewer #3: Partly

2. Has the statistical analysis been performed appropriately and rigorously? 

Reviewer #1: No

Reviewer #2: Yes

Reviewer #3: I Don't Know

3. Have the authors made all data underlying the findings in their manuscript fully available?

Reviewer #1: No

Reviewer #2: No

Reviewer #3: Yes

4. Is the manuscript presented in an intelligible fashion and written in standard English?

Reviewer #1: Yes

Reviewer #2: Yes

Reviewer #3: Yes

5. Review Comments to the Author

Reviewer #1: This manuscript addresses an important topic, and aims to study how the parameters in a cardiac model influence simulated ECG waveforms. The simulations are informed by data from patients.

Although the study is well motivated, I have three major concerns:

1. The methodology for sensitivity analysis is not clearly described, and it is unclear how the model parameters have been varied. Table 3 indicates that 7 parameters are involved, and it is not clear that this 7-dimensional parameter space has been explored evenly and completely. At the very least, I would have expected an approach such as latin hypercube sampling to be used. I would recommend that the authors substantially revise their approach in the light of both a recent paper from Saltelli -- https://doi.org/10.1016/j.envsoft.2019.01.012, and a review from the Maastricht group -- https://doi.org/10.1002/cnm.2755. Both of these papers provide definitions of uncertainty and sensitivity analysis. They give a good introduction to methods for sensitivity analysis, and recommend suitable methods for robust and systematic approaches.

2. The configuration of the ECG is known to depend on activation and recovery times in the myocardium, as well as the torso model parameters. Activation and recovery will be strongly influenced by the model parameters provided in Table 2. These parameters are assigned fixed values based on the literature, but there is no consensus on suitable values for these parameters, with estimates varying by up to an order of magnitude.

3. Action potential shape also plays an important role in gradients of potential during repolarization, and action potential shape is in turn dependent on the wider set of model parameters. The TNNP model is a reasonable choice for this study, but data from human myocytes support a more gradual repolarization profile (https://doi.org/10.1371/journal.pcbi.1002061), and this may be altered again in patients with heart failure and needing CRT. Furthermore, the ventricles begin to contract during the T wave, which alters the tissue geometry. At the very least, these factors should be considered as significant limitations to the manuscript.

Minor concerns:

There are some issues with language, for example line 133 "structured healthy hearts" should be "structurally", and lines 232 and 234 should refer to "literature" instead of "literary".

Figures and Tables need more details in the captions. For example it is not clear what P1..P6 represent (I assume patients), and the colour maps in Figures 2,3,8-13 should be clearly labelled and explained in the figure legends.

Reviewer #2: The manuscript describes a study in which heart-torso computational models were tested for accuracy and sensitivity to parameters, using electrophysiological data from six patients. I agree that more studies of this type are needed to advance the field of personalized medicine. Your section on metrics was especially well-explained.

I have a number of recommendations for improving the manuscript, many of which have to do with clarifying meanings and improving the labeling and descriptions of the figures. My suggestions are described in more detail below.

1) There are small typos throughout the manuscript. The journal website says “PLOS ONE does not copyedit accepted manuscripts,” so I strongly encourage you to proofread the manuscript very carefully. In addition to some spelling and word-usage errors, there appear to be various LaTeX coding errors. Examples: You should re-check your citations so that none are accidentally duplicated within a list. There are several places where a word like “Sec” or “section” is followed by a blank space, instead of referring me to a specific named section. There are lots of places where upside down exclamation points occur, such as page 13 and elsewhere.

2) P.2 line 35: “has shortcomings” Please give at least one example of a shortcoming here.

3) Since the introduction is somewhat long, it was hard for me to remember all the points you were making by the time I reached the end of the intro. To help counteract this, on p. 3, line 98, it would be great if you could list a few examples of the most important unresolved questions after the portion that says, “Based on these unresolved questions.” To my understanding, novelty isn’t a requirement for PLOS ONE papers, but if there are novel or unusual aspects to your work, it would be good to emphasize these somewhere inside or near this paragraph.

4) P.5, line 152. “The numbers of used BSM leads are presented in Table 1.” I wasn’t able to find these numbers in Table 1. Which column are they in?

5) P.5: In the data preprocessing section, you describe segmentation, mesh choices, and mesh refinement. It would be nice if you could include some commentary (maybe in Discussion or limitations) about how these choices may have impacted your results. I’m not asking you to run more simulations, but it would be good to have some idea of how important you think these segment or mesh choices are.

6) Eq. 1: The first and last lines of Eq 1 are written in a way that it isn’t clear to me how the grad operator should be applied. For instance, in the first line of Eq. 1 I was expecting something more like Grad. ( Sigma_i(Grad V_m + Grad phi_e)) with the extra set of outer parentheses.

7) P.6, paragraph starting near line 180: Is beta the same as beta_m? Please define phi_b and Sigma_b.

8) Fig 1 and caption: It would help to add some sentences to explain how the figure was prepared. I don’t understand the sentence “Blue circles present …” I think the word “present” should probably be something else? What are the black dots in the figure? Do the rainbow colors in the boxes mean anything?

9) Fig 3, Fig 6-12: Please explain color-coding of curves (red vs black or blue vs. green) and/or include a legend.

10) Fig 4: I can’t see any squares in the plot, only lines with circles. I downloaded the .tif version of the figure and zoomed in and still couldn’t see any squares. Please make your squares and circles larger or more distinctive looking.

11) P.13, line 440: Is p < 0.03 still for the Mann-Whitney test? If so, you should state this.

12) P.14: Figure 7 should be introduced and described at least briefly somewhere on this page, in addition to just including a caption.

13) Fig 8-13: Include explanations in the main text of the paper to say why you’re only showing data from a subset of the patients. For example, please explain why Fig 8 only shows P1-P3. I’m not saying you need to include every patient in every plot, just explain why you showed the ones you showed.

14) P.15, line 522-524: I can’t tell which of the patients in Fig 11 belong to the “7 of 10” cases. Please list which patients belong to which category.

15) P.15, similar to before, Figure 12 should be introduced and described briefly in the text where it first appears.

16) Style notes: While there is nothing technically wrong with one-sentence paragraphs, I find them strange and wonder (especially in the conclusion) whether you can consolidate your sentences more, though if the one-sentence paragraphs were a deliberate choice that you want to retain, that’s okay.

Thanks for the time spent producing the manuscript. I can review a revised version if it becomes available.

Reviewer #3: The Abstract begins: “The objectives of this study were to evaluate the accuracy of personalized numerical simulations…” I do not think the results accomplish that objective. For example, “We found a comparatively good correlation (r > 0:72) between the simulated and real ECG for 8 of 10 cases (80%).” First, r > 0.72 seems arbitrary. In Fig. 1 the average r > 0.72 for most data sets, but there is a tremendous variability among recording sites, including one (or many) negative correlations in all 10 datasets. What does this mean? How good an agreement between model and data would be needed for it to be clinically useful? A correlation coefficient may be unbiased, but that is an advantage and a disadvantage: a point-by-point summary of error may or may not highlight relevant features. For example, in Fig. 3 it is noted that the polarity of the QRS complexes are sometimes inverted between data and model; isn’t that a sign that the model is badly wrong? In general, the value of a model depends on how it is to be used, but this is not critically evaluated here. The study does not draw strong conclusions about the value of such personalized models, but the conclusions that are drawn do not seem to be well supported.

The English is generally OK, with some issues: The sentence beginning on line 53 is awkward. Lines 232, 234 "literature" not "literary" values.

6. PLOS authors have the option to publish the peer review history of their article (what does this mean?). If published, this will include your full peer review and any attached files.

Reviewer #1: No

Reviewer #2: No

Reviewer #3: No

---

## [Author Response · Author response to Decision Letter 0]

26 Aug 2020

We thank all reviewers for their meaningful comments. All comments were taken into account and the corresponding changes were made to the text.

Reviewer #1: This manuscript addresses an important topic, and aims to study how the parameters in a cardiac model influence simulated ECG waveforms. The simulations are informed by data from patients.

Although the study is well motivated, I have three major concerns:

1. The methodology for sensitivity analysis is not clearly described, and it is unclear how the model parameters have been varied. Table 3 indicates that 7 parameters are involved, and it is not clear that this 7-dimensional parameter space has been explored evenly and completely. At the very least, I would have expected an approach such as latin hypercube sampling to be used. I would recommend that the authors substantially revise their approach in the light of both a recent paper from Saltelli -- https://doi.org/10.1016/j.envsoft.2019.01.012, and a review from the Maastricht group -- https://doi.org/10.1002/cnm.2755. Both of these papers provide definitions of uncertainty and sensitivity analysis. They give a good introduction to methods for sensitivity analysis, and recommend suitable methods for robust and systematic approaches.

RESPONSE: Thank you very much for the review. Replying to the above note, we would like to clarify the approaches we used in this article. 

Indeed, we have not used here conventional sensitivity analysis approaches reported in the papers you mentioned. We focused on the different tasks in this study. 

First, we compared experimental data recorded in 10 clinical cases for 6 patients at different ventricular activation protocols with simulations computed using the reference model with population-based parameters conventionally used in many modeling studies by different author groups. The model simulations were first compared with experimental data in terms of qualitative correlation metrics. The analysis reveals the different quality of model predictions for different patients and at different ECG lead locations suggesting that model parameters should be tailored to the personal patient data. The main conclusion from this part of the study is that the correlation metrics is a good estimate for the quality of the reference simulations and the identification problem statement. Next point is that even adequate ECG simulation for 12 standard leads cannot guarantee the quality of simulations in the entire body. 

The results obtained in the first part of the study raises the question of the justification and adequacy of a parameter identification problem in a physiologically non-implausible range for several chosen parameters. This task was addressed in a simple possible way of analysis using one-by-one parameter variation in the physiological range with fixed other parameters. Here, each free parameter was tested in the physiological range and a relative Euclidean distance (RED) between the simulated and patient ECG was calculated as a function upon the variable parameter value. By plotting such a function, we can see if the minimum for the distance lays within or at the border of the permissive interval. The function visualizes how much the RED changes between the reference and minimum parameter value and how far the minimum is from the reference. Such analysis shows if this parameter variation may essentially contribute to a reduction in the model discrepancy with the experimental data compared to the reference model. The main conclusion from this analysis was that the best parameter values for the minimal RED were more frequently located at the edge of the physiological interval, suggesting more wide permissive intervals for solving the identification problem. We also found a shortlist of parameters essentially contributing to ECG that could be varied to perform model tailoring to experimental data. These parameters were myocardial anisotropy and conductivities of some internal organs.

These predictions were then double-checked in the following analysis of the model sensitivity to variation of the parameters with respect to the reference model. Here, we analyzed if the change in the parameter causes a significant effect on the model simulations as compared to the reference values. The approach for analysis we used in this article was adapted from Keller et.al, 2012 for signal comparison and from Sanchez et. al, 2018 for physiological biomarker comparison. We used integrative (aggregated) relative Euclidean distance between model signals (transmembrane potential, extracellular potential) throughout some myocardial area (ventricular surfaces, entire myocardial volume, regional locations on myocardial surfaces, torso, ECG lead locations) as a measure of the model proximity to the reference outputs. We computed the maximal value of such distances within the physiological intervals for the parameter variation and analyzed if the effect of parameter change is physiologically significant. The main findings of this analysis were that different parameters demonstrate an essential contribution to different model output signals and find 2-3 parameters that variation has the significant impact on either the transmural potential or on the extracellular potential, and on the physiological characteristics of the signals (e.g. amplitudes, durations of ECG complexes, action potential duration). For the first time, we showed regional sensitivity of the myocardial model to parameter variation which then can be used for solving model tailoring tasks.

So, we understand that our uni-parametric model analysis is the first step for further multi-parametric model analysis using combinations of varied parameters in the multi-dimensional parametric space, but we believe that our results allow one to limit the number of free parameters to reduce the complexity of the identification problem.

We have edited the Model analysis section in the manuscript to make clear our approaches and metrics to assess the discrepancy between simulated and experimental data and distinctions between model simulations at varying parameters (see p. 7 in the manuscript). A passage on further application of conventional methods for sensitivity analysis is added into the Limitations (pp. 22) and Discussion (see p. 23. line 668).

2. The configuration of the ECG is known to depend on activation and recovery times in the myocardium, as well as the torso model parameters. Activation and recovery will be strongly influenced by the model parameters provided in Table 2. These parameters are assigned fixed values based on the literature, but there is no consensus on suitable values for these parameters, with estimates varying by up to an order of magnitude.

RESPONSE: We agree that there is no consensus about a number of model parameters, e.g. surface-to-volume ratio and the membrane capacitance indicated in Table 2. That is why we choose these values from computer modeling that works with good simulation results. We added a notice about that in Methods (p. 8, line 286) and Limitations (p. 27) sections indicating the choice of the parameters used in the study.

3. Action potential shape also plays an important role in gradients of potential during repolarization, and action potential shape is in turn dependent on the wider set of model parameters. The TNNP model is a reasonable choice for this study, but data from human myocytes support a more gradual repolarization profile (https://doi.org/10.1371/journal.pcbi.1002061), and this may be altered again in patients with heart failure and needing CRT. Furthermore, the ventricles begin to contract during the T wave, which alters the tissue geometry. At the very least, these factors should be considered as significant limitations to the manuscript.

RESPONSE: We agree that the use of different cellular models would additionally support our predictions. OHR2011 is a perfect model that reproduces a lot of experimental data, but this model is more consuming in terms of computational time. For our computationally intense study using the bi-domain model, we chose TNNP2006 as it combines an acceptable accuracy of simulations for cardiomyocyte electrophysiology with the lowest requirements for computational resources.

We agree that accounting for cellular remodeling at heart failure is challenging for future work. The effects of mechanics on the excitation is also a problem which has to be addressed in the future analysis. We added notes on these limitations in the Limitations. (p. 27)

Minor concerns:

There are some issues with language, for example line 133 "structured healthy hearts" should be "structurally", and lines 232 and 234 should refer to "literature" instead of "literary".

RESPONSE: This was corrected.

Figures and Tables need more details in the captions. For example it is not clear what P1..P6 represent (I assume patients), and the colour maps in Figures 2,3,8-13 should be clearly labelled and explained in the figure legends.

RESPONSE: This was edited.

Reviewer #2: The manuscript describes a study in which heart-torso computational models were tested for accuracy and sensitivity to parameters, using electrophysiological data from six patients. I agree that more studies of this type are needed to advance the field of personalized medicine. Your section on metrics was especially well-explained.

I have a number of recommendations for improving the manuscript, many of which have to do with clarifying meanings and improving the labeling and descriptions of the figures. My suggestions are described in more detail below.

RESPONSE: We thank the reviewer for valuable concerns on our manuscript.

1) There are small typos throughout the manuscript. The journal website says “PLOS ONE does not copyedit accepted manuscripts,” so I strongly encourage you to proofread the manuscript very carefully. In addition to some spelling and word-usage errors, there appear to be various LaTeX coding errors. Examples: You should re-check your citations so that none are accidentally duplicated within a list. There are several places where a word like “Sec” or “section” is followed by a blank space, instead of referring me to a specific named section. There are lots of places where upside down exclamation points occur, such as page 13 and elsewhere.

RESPONSE: This was edited.

2) P.2 line 35: “has shortcomings” Please give at least one example of a shortcoming here.

RESPONCE: We edited and clarified this paragraph (p. 2, line 59-69)

3) Since the introduction is somewhat long, it was hard for me to remember all the points you were making by the time I reached the end of the intro. To help counteract this, on p. 3, line 98, it would be great if you could list a few examples of the most important unresolved questions after the portion that says, “Based on these unresolved questions.” To my understanding, novelty isn’t a requirement for PLOS ONE papers, but if there are novel or unusual aspects to your work, it would be good to emphasize these somewhere inside or near this paragraph.

RESPONSE: We agree with your comments. The Introduction section is essentially rewritten (p. 1-4).

4) P.5, line 152. “The numbers of used BSM leads are presented in Table 1.” I wasn’t able to find these numbers in Table 1. Which column are they in?

RESPONSE: Sorry about that. We did not show this information in the article. Not more than 6.25% of 224 electrodes were excluded from consideration, because of poor connection with the body. This is mentioned on page 5, lines 171-174.

5) P.5: In the data preprocessing section, you describe segmentation, mesh choices, and mesh refinement. It would be nice if you could include some commentary (maybe in Discussion or limitations) about how these choices may have impacted your results. I’m not asking you to run more simulations, but it would be good to have some idea of how important you think these segment or mesh choices are.

RESPONSE: We added a new subsection “Reasons of model shortcomings” to the Discussion section (p. 27)

6) Eq. 1: The first and last lines of Eq 1 are written in a way that it isn’t clear to me how the grad operator should be applied. For instance, in the first line of Eq. 1 I was expecting something more like Grad. ( Sigma_i(Grad V_m + Grad phi_e)) with the extra set of outer parentheses.

RESPONSE: We edited the equations (p. 5, line 202, eq. 1).

7) P.6, paragraph starting near line 180: Is beta the same as beta_m? Please define phi_b and Sigma_b.

RESPONSE: This was edited (pp. 6, line 203-216).

8) Fig 1 and caption: It would help to add some sentences to explain how the figure was prepared. I don’t understand the sentence “Blue circles present …” I think the word “present” should probably be something else? What are the black dots in the figure? Do the rainbow colors in the boxes mean anything?

RESPONSE: We edited figure caption and legend (p. 10, Fig. 1). The colors of the bars have no meaning. Other marks have been clarified.

9) Fig 3, Fig 6-12: Please explain color-coding of curves (red vs black or blue vs. green) and/or include a legend.

RESPONSE: Definitions of color-coding in figures are added (see Figs. 1-13). 

10) Fig 4: I can’t see any squares in the plot, only lines with circles. I downloaded the .tif version of the figure and zoomed in and still couldn’t see any squares. Please make your squares and circles larger or more distinctive looking.

RESPONSE: We changed Fig. 4 and the description (p. 14).

11) P.13, line 440: Is p < 0.03 still for the Mann-Whitney test? If so, you should state this.

RESPONSE: We have mentioned using the Mann-Whitney test where appropriate in the text. We used this non-parametric test because of the small number of observations and the non-Gaussian distribution of the variables.

12) P.14: Figure 7 should be introduced and described at least briefly somewhere on this page, in addition to just including a caption.

RESPONSE: We have added references to Fig. 6 and 7 in the text (p. 16, lines 707,729).

13) Fig 8-13: Include explanations in the main text of the paper to say why you’re only showing data from a subset of the patients. For example, please explain why Fig 8 only shows P1-P3. I’m not saying you need to include every patient in every plot, just explain why you showed the ones you showed.

RESPONSE: We showed data from some patient models as representative examples of described spatial patterns of the regional sensitivity (in terms of RED) map on the myocardial surface or torso surface. For the rest of the models, the patterns are similar if not specifically indicated and not shown. 

14) P.15, line 522-524: I can’t tell which of the patients in Fig 11 belong to the “7 of 10” cases. Please list which patients belong to which category.

RESPONSE: We added a list of models with an unstructured pattern. (pp 19, lines 560-562)

Further, regions with a high effect of the endo/epi ratio and blood conductivity variation are co-localized in the late activation zone of the ventricles in 7 of 10 cases except for P4(RV1), P4(RV2) and P5(LV).

15) P.15, similar to before, Figure 12 should be introduced and described briefly in the text where it first appears.

RESPONSE: We added the description of the Figure (pp 19. lines 565-568). Please, note that the order of Figures 12 and 13 in the new text is reversed.

16) Style notes: While there is nothing technically wrong with one-sentence paragraphs, I find them strange and wonder (especially in the conclusion) whether you can consolidate your sentences more, though if the one-sentence paragraphs were a deliberate choice that you want to retain, that’s okay.

RESPONSE: We merged one-sentence paragraphs.

Thanks for the time spent producing the manuscript. I can review a revised version if it becomes available.

Reviewer #3: The Abstract begins: “The objectives of this study were to evaluate the accuracy of personalized numerical simulations…” I do not think the results accomplish that objective. For example, “We found a comparatively good correlation (r > 0:72) between the simulated and real ECG for 8 of 10 cases (80%).” First, r > 0.72 seems arbitrary. In Fig. 1 the average r > 0.72 for most data sets, but there is a tremendous variability among recording sites, including one (or many) negative correlations in all 10 datasets. What does this mean? How good an agreement between model and data would be needed for it to be clinically useful? A correlation coefficient may be unbiased, but that is an advantage and a disadvantage: a point-by-point summary of error may or may not highlight relevant features. For example, in Fig. 3 it is noted that the polarity of the QRS complexes are sometimes inverted between data and model; isn’t that a sign that the model is badly wrong? In general, the value of a model depends on how it is to be used, but this is not critically evaluated here. The study does not draw strong conclusions about the value of such personalized models, but the conclusions that are drawn do not seem to be well supported.

RESPONSE: We thank the reviewer for valuable concerns on our manuscript. One of the main objectives of this study was to evaluate the accuracy of personalized numerical simulations of the electrical activity in human ventricles by comparing simulated electrocardiogram (ECG) with real patients’ ECG.

Your major comment on the manuscript was that the results do not accomplish that objective for the following reasons. First, the correlation coefficient between real and reconstructed ECGs seems to be an insufficiently reliable indicator of the quality of modeling accuracy. Second, the article does not critically analyze the results of modeling accuracy and does not make a strong conclusion about the clinical value of this type of model.

We agree with your appraisal, but only partially. The advantages and disadvantages of the correlation coefficient were briefly discussed in the section Methods (p. 8, line 287-297). In particular, we noticed that this metric is weakly sensitive to small changes in signal values and variations in signal amplitudes. However, as we have noted, the correlation coefficient is a generally accepted measure of the qualitative difference between signals and has been used in several studies with a similar design (lines 289-293). The correlation coefficient allows us to compare the quality of our simulation with the published data. This is why we have limited ourselves to providing accurate analysis results in terms of correlation. We choose the level of the mean correlation coefficient above 0.7 as a measure of the strong correlation which is widely agreed in the statistical applications in medicine (Akoglu et.al, 2018). Eight model cases over perform this threshold. Three cases of simulation (P1(LV), P1(RV), and P6(RV)) overperform an 80% level of high accuracy suggested in the other works, e.g. see Keller et. al, 2011. These cases are classified into the first group of high correlation models with CC>0.80 as described in the text (p. 10, line 330).

However, we agree with you that more quantitative and clinically interpretive estimates of model accuracy would add information content for the results presented in the manuscript. In this regard, we added an analysis using another metric, the normalized mean-square deviation in the revised version of the manuscript. These metrics are also often used to compare bioelectric signals. Accordingly, we added the definition of this metrics to the Methods section and presented the comparison results in terms of this metrics in “Results / Comparison of reference simulations and PECG (p. 10)” section and their discussion in Discussion/ Accuracy of the ECG simulation section (p. 20). These metrics were used in addition to the relative Euclidean distance between simulated and patient data (see the section Model analysis ) which is also a kind of quantitative estimate of the model quality. This was used to assess the possibility to use uni-parametric optimization of the model which allows us to predict the most important parameters affecting the proximity of simulations to the experimental data.

Despite the relatively high average correlation between simulated and real ECGs, which was found in the numerical tests, there was a high level of error for certain models. In the last version of the manuscript, we concluded that the models with the averaged population-based parameter values do not allow us to provide personal ECG modeling for any cases (section “Conclusions”, lines 830 - 836). In the revised version of the manuscript, we further highlighted the negative results about the lack of accuracy and strong variability in the simulation results. In particular, we added data on the variability of the correlation coefficient to the abstract and drew attention to the negative correlation coefficients and the opposite polarity of QRS complexes in some ECG leads in the “Discussion” section. We also noted once again that the clinical application of the models with the averaged population-based values parameter has significant limitations.

The English is generally OK, with some issues: The sentence beginning on line 53 is awkward. Lines 232, 234 "literature" not "literary" values.

RESPONSE: This was edited in the text.

---

## [Decision Letter · Decision Letter 1]

29 Sep 2020

PONE-D-20-04786R1

Parameter variations in personalized electrophysiological models of human heart ventricles

PLOS ONE

Dear Dr. Ushenin,

Thank you for submitting your manuscript to PLOS ONE. After careful consideration, we feel that it has merit but does not fully meet PLOS ONE’s publication criteria as it currently stands. Therefore, we invite you to submit a revised version of the manuscript that addresses the points raised during the review process.

Please address all comments indicated by the Reviewers.

We look forward to receiving your revised manuscript.

Kind regards,

Elena G. Tolkacheva, PhD

Academic Editor

PLOS ONE

Reviewers' comments:

Reviewer's Responses to Questions

**Comments to the Author**

1. If the authors have adequately addressed your comments raised in a previous round of review and you feel that this manuscript is now acceptable for publication, you may indicate that here to bypass the “Comments to the Author” section, enter your conflict of interest statement in the “Confidential to Editor” section, and submit your "Accept" recommendation.

Reviewer #1: (No Response)

Reviewer #2: (No Response)

Reviewer #3: (No Response)

2. Is the manuscript technically sound, and do the data support the conclusions?

Reviewer #1: Partly

Reviewer #2: Yes

Reviewer #3: Yes

3. Has the statistical analysis been performed appropriately and rigorously? 

Reviewer #1: No

Reviewer #2: Yes

Reviewer #3: I Don't Know

4. Have the authors made all data underlying the findings in their manuscript fully available?

Reviewer #1: No

Reviewer #2: No

Reviewer #3: Yes

5. Is the manuscript presented in an intelligible fashion and written in standard English?

Reviewer #1: No

Reviewer #2: No

Reviewer #3: Yes

6. Review Comments to the Author

Reviewer #1: The revision addresses some of my concerns. However concern 2 in my original review remains.

I wrote "2. The configuration of the ECG is known to depend on activation and recovery times in the myocardium, as well as the torso model parameters. Activation and recovery will be strongly influenced by the model parameters provided in Table 2. These parameters are assigned fixed values based on the literature, but there is no consensus on suitable values for these parameters, with estimates varying by up to an order of magnitude." I believe that this point underlies the poor correlations between some of the simulated and recorded ECG signals (e.g correlation coefficients of -1 in Figure 1). I would consider other measures with which to compare the simulated and recorded ECGs: QRS duration, QT interval, Twave symmetry are all metrics that might provide valuable information.

I understand the authors' response to my comment about sensitivity analysis. I would like to see a rationale for the analysis of parameters in Table 3. Why include these and not others?

There remain some problems with the English, and the manuscript must be checked by a native English speaker.

For example:

Line 265 'propose' -> 'purpose'

Line 286 'includes' -> 'include'

Line 293 'metrics is' -> 'metric is' or 'metrics are'

Table 4 'electode' -> 'electrode'

Reviewer #2: Thank you for revising the manuscript and for your responses to my comments. I have a major comment and a minor comment about the revised version.

Major comment: My main remaining issue is that my previous advice (to proofread the manuscript very carefully) wasn’t addressed as thoroughly as I’d hoped. I appreciate that the authors fixed typos that I explicitly pointed out, but what I was trying to convey in my first review is that there were lots of other mistakes that I didn’t have time to list individually. Although I understand that some typos are inevitable in a manuscript of this length, the number of mistakes is still higher than what I’m used to seeing. I encourage the authors to find a native English speaker (or as close as they can find to one) to read the manuscript carefully, though some other proofreading approach would be needed to handle issues with figures and equations. Here are some examples of problems I encountered:

- I can’t read the numbers along the axes in certain figures. For example, the axis numbers for the blue and green ECG curve plots in Figures 9-11 are especially hard to read. The figures are very small and blurry. Downloading the .tif files didn’t help clarify anything. Perhaps the figures look good on your side, but I can’t see them clearly. Were you able to download the version supplied to the publisher and check all figures for clarity?

- There are typos or formatting errors in several of the equations that were added to the revised version, including but not limited to Eq. 11.

- There are various grammatical and word-usage errors. For example, there are cases where a word is spelled correctly but the wrong word is being used, such as in the caption of Figure 4.

There are many other mistakes besides the ones I pointed out above. I don’t think it should be my role to fully edit the manuscript, and I don’t think PLoS-ONE will do it, so I’m not sure how to proceed. If the journal’s policy is that they don’t provide copy-editing, that puts more of the proofreading burden on the authors compared with some other journals.

Minor comment: Please clarify what types of mesh nodes are included in set “I” for Eqs. 12, 13, 15. I can find an explanation of what “I” is where Equation 14 is introduced but I don’t know if that description of “I” pertains to the other equations as well. It’s possible this was addressed somewhere and I missed it. It would be fine to define the sets later when relevant tables or figures are introduced in Results, if that is easier.

Reviewer #3: The authors have addressed my concerns. However, I remain unsure about how such personalized models would be used clinically, and what level of agreement between model and data is needed for the model to be useful.

7. PLOS authors have the option to publish the peer review history of their article (what does this mean?). If published, this will include your full peer review and any attached files.

Reviewer #1: No

Reviewer #2: No

Reviewer #3: No

---

## [Author Response · Author response to Decision Letter 1]

20 Nov 2020

We thank all reviewers for their meaningful comments. All comments were taken into account and the corresponding changes were made to the text.

Reviewer #1: The revision addresses some of my concerns. However concern 2 in my original review remains.

I wrote "2. The configuration of the ECG is known to depend on activation and recovery times in the myocardium, as well as the torso model parameters. Activation and recovery will be strongly influenced by the model parameters provided in Table 2. These parameters are assigned fixed values based on the literature, but there is no consensus on suitable values for these parameters, with estimates varying by up to an order of magnitude." I believe that this point underlies the poor correlations between some of the simulated and recorded ECG signals (e.g correlation coefficients of -1 in Figure 1). I would consider other measures with which to compare the simulated and recorded ECGs: QRS duration, QT interval, Twave symmetry are all metrics that might provide valuable information.

I understand the authors' response to my comment about sensitivity analysis. I would like to see a rationale for the analysis of parameters in Table 3. Why include these and not others?

RESPONSE: Your comments show that two points are not clearly stated in the manuscript. The first question is about what the values of the model parameters were used for the reference model and why. The second question is what parameters were selected for variation in the frame of the sensitivity analysis and what was the motivation for this selection. Let us try to give detailed answers to these questions.

The parameters of the model can be divided into three groups. The first group consists of parameters of the cellular model, i.e. parameters of ionic currents in cardiomyocytes. The second group includes parameters of the bidomain model: cell membrane capacity, the surface-to-volume ratio of the cardiomyocytes, and coefficients of the conductivity tensors of myocardial tissue. The third group of model parameters includes parameters of the torso organ conductivities. In addition, each group can be extended with parameters that characterize the spatial heterogeneity of the parameters from this group.

The reference model is a model that we used to compare the simulation results with the ECG of patients and as a reference point for sensitivity analysis. For the reference model, the parameter values were assigned as follows. The parameter values of the first group were taken from the original work (ten Tusscher et. al. 2006), since we assume that TNNP 2006 model of human ventricular cardiomyocytes has an optimal balance of complexity and adequacy. Moreover, we took into account the transmural and apicobasal heterogeneity of the potassium currents (g_Ks, g_to). A significant influence of these parameters on ECG morphology was shown in the work (Keller et. al, 2012). 

In our work we used population-based values for the parameters of the second group, based on the previous modeling works (Keller et. al, 2010, Keller et. al, 2012, Boulakia et. al., 2010, Sánchez et. al., 2018), which carefully selected a plausible range of values to simulate adequate characteristics of ECG recorded in patients. In particular, these parameter values allow the model to produce: (1) a realistic conduction velocity 0.5-0.6 m/s in myofiber direction and 0.15-0.25 m/s across the fibers which is reported in literature, (2) to simulate QRS width higher than 100 ms that is close to patient recordings upon the point stimulation.

We understand that these parameters may be patient-specific and vary in different regions of the myocardium, so our assumption should be considered as a limitation of the study. However, variation in the parameters would change the conduction velocity, but not the direction of the wave-front so the choice of the parameters cannot induce wave inversion and essentially improve the quality of simulations in cases where correlations between ECG recordings and simulations are rather poor. In contrast, the local heterogeneity in the conduction velocity can essentially affect the ECG morphology, but this issue was out of the scope of this study.

The third group of parameters included electrical conductivity values of the lungs, blood in the heart cavity and large vessels, liver, and bone tissue of the spine. Thus, all significant electrical conductivity heterogeneities were taken into account in our model with the exception of pericardial fat, ribs, and sternum. We also used population-based values for parameters that have been reported in previous works Keller et. al, 2010, Sánchez et. al., 2018.

When selecting model parameters for variation for the purpose of sensitivity analysis (see Table 3), we focused on those parameters whose effects were not analyzed in detail in previous studies. We intentionally did not analyze the effects of the ionic parameters on simulated ECG as model sensitivity to their variation and effects on ECG characteristics were previously studied in several works in detail (Keller et. al, 2010). However, coefficients of epi/endo and apicobasal heterogeneity was included in the analysis because of the fact that their essential role in the T-wave orientation was previously demonstrated in the model at the normal activation sequence, while no data was available on their significance for ECG morphology in the cases of ectopic action.

It is known that the ratio of cell membrane capacity to the “surface-to-volume ratio” affects strongly the conduction velocity. In our work, we have tried to maintain a realistic conduction velocity for all simulation cases, so we did not vary the membrane capacitance and the surface-to-volume ratio.

When varying the parameters of anisotropic electrical conductivity of the myocardium, we used a special approach. It is known that there is a correlation between the coefficients of anisotropic electrical conductivity of the myocardium. We used Roth et. al. mathematical framework that establishes a relationship between extracellular and intracellular conductivities. This approach allows us to vary the values of the four electrical conductivity parameters, based on varying the value of only one parameter, the so-called anisotropy ratio ($\\lambda_L/\\lambda_T$). We used a variation of this parameter instead of variations of the four parameters for our sensitivity analysis.

Finally, we varied the conductivity of the internal organs and blood intending, first of all, to study the regional effects of these variations. In particular, we hypothesized the regional effects of variations of spine conductivity on the ECG characteristics. However, modeling results did not support this hypothesis as discussed in the paper.

Thanks to your comments, we have seen that these important points are not clearly stated in the manuscript. In particular, the caption to table two contains an incorrect statement.

We have made appropriate corrections in the "methods" and "discussion" sections and in the Tables 2 and 3.

Answering your suggestion on using ECG features to estimate the effects of parameter variation, we provided such analysis as reported in the Section “Effects of parameter variation on properties of myocardial depolarization and repolarization”. We analyzed the effects of parameter variations on the amplitude and time-to-peak values of QRS complexes and T-waves, which can be accurately detected by ECG signals. However, we did not use such parameters as QT-interval, QRS duration, J-point, T-wave asymmetry when comparing the simulation results with the patients' ECGs. We understand these ECG characteristics are often used in clinical practice. The problem is that their clinical value was investigated only for sinus rhythm. Their applicability in the case of ectopic heartbeats originated from various sites of the ventricles is not clear. Moreover, detecting T-wave boundaries and measuring the QT-interval is challenging for ectopic ventricular contractions.

There remain some problems with the English, and the manuscript must be checked by a native English speaker.

For example:

Line 265 'propose' -> 'purpose'

Line 286 'includes' -> 'include'

Line 293 'metrics is' -> 'metric is' or 'metrics are'

Table 4 'electode' -> 'electrode'

RESPONSE: All noted mistakes were fixed. Additionally, we used an external proofreading service to fix as many mistakes as possible.

Reviewer #2: Thank you for revising the manuscript and for your responses to my comments. I have a major comment and a minor comment about the revised version.

Major comment: My main remaining issue is that my previous advice (to proofread the manuscript very carefully) wasn’t addressed as thoroughly as I’d hoped. I appreciate that the authors fixed typos that I explicitly pointed out, but what I was trying to convey in my first review is that there were lots of other mistakes that I didn’t have time to list individually. Although I understand that some typos are inevitable in a manuscript of this length, the number of mistakes is still higher than what I’m used to seeing. I encourage the authors to find a native English speaker (or as close as they can find to one) to read the manuscript carefully, though some other proofreading approach would be needed to handle issues with figures and equations. Here are some examples of problems I encountered:

RESPONSE: We apologize for the quality of the previous text. We have used a proofreading service for the current version.

- I can’t read the numbers along the axes in certain figures. For example, the axis numbers for the blue and green ECG curve plots in Figures 9-11 are especially hard to read. The figures are very small and blurry. Downloading the .tif files didn’t help clarify anything. Perhaps the figures look good on your side, but I can’t see them clearly. Were you able to download the version supplied to the publisher and check all figures for clarity?

RESPONSE: We have improved quality of images for Figures 3,4, 9-11.

- There are typos or formatting errors in several of the equations that were added to the revised version, including but not limited to Eq. 11.

RESPONSE: Typos in Equation 11 and some equations in the text were fixed.

- There are various grammatical and word-usage errors. For example, there are cases where a word is spelled correctly but the wrong word is being used, such as in the caption of Figure 4.

RESPONSE: We have fixed word-usage errors in the caption of Figure 4,5 and some other places in the text.

There are many other mistakes besides the ones I pointed out above. I don’t think it should be my role to fully edit the manuscript, and I don’t think PLoS-ONE will do it, so I’m not sure how to proceed. If the journal’s policy is that they don’t provide copy-editing, that puts more of the proofreading burden on the authors compared with some other journals.

RESPONSE: We used an external proofreading service to fix as many mistakes as possible.

Minor comment: Please clarify what types of mesh nodes are included in set “I” for Eqs. 12, 13, 15. I can find an explanation of what “I” is where Equation 14 is introduced but I don’t know if that description of “I” pertains to the other equations as well. It’s possible this was addressed somewhere and I missed it. It would be fine to define the sets later when relevant tables or figures are introduced in Results, if that is easier.

RESPONSE: Done.

Reviewer #3: The authors have addressed my concerns. However, I remain unsure about how such personalized models would be used clinically, and what level of agreement between model and data is needed for the model to be useful.

RESPONSE: A brief overview of possible clinical applications of personal computer models of cardiac electrical activity is given in the Introduction section of this manuscript. In particular, personal models can be useful in planning catheter ablation of cardiac arrhythmias, optimizing cardiac resynchronization therapy, and stratifying the risk of sudden cardiac death in patients with ischemic and non-ischemic cardiomyopathies, etc. (see references [2]-[8]).

What level of agreement between model output and data is needed for the model to be useful in clinics? Theoretically, modeling errors should not exceed the typical ECG recording error (electrical and muscle noise, changes in the contact resistance of the ECG electrode with the skin, inaccuracies in determining the positions of the standard ECG leads (V1-V6), etc.) and the level of ECG variability caused, for example, by the patient's breathing. However, each medical application requires its own level of modeling accuracy, which must be identified through clinical research. Nevertheless, we may assume that 4 out of 10 cases of our simulation provided accuracy close to that required for clinical applications. On the one hand, this shows that the required level of model accuracy is fundamentally achievable. On the other hand, modeling using averaged population-based parameters does not work well and this approach cannot be directly translated into clinical practice.

We have added a brief comment on the required modeling accuracy for clinical applications in the discussion section of this manuscript (page 24, Accuracy of the ECG simulation).

---

## [Decision Letter · Decision Letter 2]

23 Dec 2020

PONE-D-20-04786R2

Parameter variations in personalized electrophysiological models of human heart ventricles

PLOS ONE

Dear Dr. Ushenin,

Thank you for submitting your manuscript to PLOS ONE. After careful consideration, we feel that it has merit but does not fully meet PLOS ONE’s publication criteria as it currently stands. Therefore, we invite you to submit a revised version of the manuscript that addresses the points raised during the review process.

Please address minor comments from the reviewer.

We look forward to receiving your revised manuscript.

Kind regards,

Elena G. Tolkacheva, PhD

Academic Editor

PLOS ONE

Reviewers' comments:

Reviewer's Responses to Questions

**Comments to the Author**

1. If the authors have adequately addressed your comments raised in a previous round of review and you feel that this manuscript is now acceptable for publication, you may indicate that here to bypass the “Comments to the Author” section, enter your conflict of interest statement in the “Confidential to Editor” section, and submit your "Accept" recommendation.

Reviewer #2: (No Response)

2. Is the manuscript technically sound, and do the data support the conclusions?

Reviewer #2: Yes

3. Has the statistical analysis been performed appropriately and rigorously? 

Reviewer #2: I Don't Know

4. Have the authors made all data underlying the findings in their manuscript fully available?

Reviewer #2: No

5. Is the manuscript presented in an intelligible fashion and written in standard English?

Reviewer #2: No

6. Review Comments to the Author

Reviewer #2: Thank you for addressing my comments. I appreciate that you hired a proofreading service and that you boosted font sizes on certain figures.

However, the level of care taken in checking the figures still falls short of what I was expecting, since it appears that new figure-related mistakes were introduced in this draft, and in at least one case a mistake persisted from the previous draft. Steps that I think your team (or the proofreading service) should always carry out when submitting a manuscript are to (1) compare every figure caption to the version of the figure that you uploaded. Are the number of elements described in the caption the same as the number shown in the figure? (2) Compare every figure with its counterpart from the previous draft. If there are any significant changes to the figure format (aside from just editing for clarity), determine whether these changes were intentional, and if so, explain the rationale for the changes in your response to the reviewers.

I’d prefer to focus my efforts on reviewing technical content rather than manuscript format. If for some reason I’m the one who’s looking at the wrong files, I apologize, but otherwise I find it hard to build trust in the rest of the manuscript, or to determine whether the figures support the conclusions, if the steps I described previously aren’t followed consistently.

7. PLOS authors have the option to publish the peer review history of their article (what does this mean?). If published, this will include your full peer review and any attached files.

Reviewer #2: No

---

## [Author Response · Author response to Decision Letter 2]

7 Feb 2021

We thank the reviewer for meaningful comments and especially for attention to technical details and errors in the figures of the manuscript.

Reviewer #2: Thank you for addressing my comments. I appreciate that you hired a proofreading service and that you boosted font sizes on certain figures.

However, the level of care taken in checking the figures still falls short of what I was expecting, since it appears that new figure-related mistakes were introduced in this draft, and in at least one case a mistake persisted from the previous draft. Steps that I think your team (or the proofreading service) should always carry out when submitting a manuscript are to (1) compare every figure caption to the version of the figure that you uploaded. Are the number of elements described in the caption the same as the number shown in the figure? (2) Compare every figure with its counterpart from the previous draft. If there are any significant changes to the figure format (aside from just editing for clarity), determine whether these changes were intentional, and if so, explain the rationale for the changes in your response to the reviewers.

I’d prefer to focus my efforts on reviewing technical content rather than manuscript format. If for some reason I’m the one who’s looking at the wrong files, I apologize, but otherwise I find it hard to build trust in the rest of the manuscript, or to determine whether the figures support the conclusions, if the steps I described previously aren’t followed consistently.

RESPONSE: Unfortunately, we did not notice some errors in the final version of the manuscript, which was automatically generated by the journal's submission system, and we are sorry for that. In particular, Figure 1 was not replaced and remained the same as in the previous version. The same image was used for Figures 12 and 13. These errors occurred at the stage of uploading the manuscript on the journal's website.

We carefully checked all the drawings and figure captions in the automatically generated document and made the necessary corrections.

Furthermore, we once again checked all the numerical values in the text, tables, and figure captions. We fixed the numerical values on page 12 (lines 411-419) and page 15 (lines 512-529). We also presented the numerical data on pages 16 (lines 546-558) in a ‘mean+std’ format. A caption in Figure 3 was corrected according to the presented figure.

We once again used a proofreading service and corrected the grammar, spelling, and punctuation of the text as suggested by the editor.

---

## [Decision Letter · Decision Letter 3]

11 Mar 2021

Parameter variations in personalized electrophysiological models of human heart ventricles

PONE-D-20-04786R3

Dear Dr. Ushenin,

We’re pleased to inform you that your manuscript has been judged scientifically suitable for publication and will be formally accepted for publication once it meets all outstanding technical requirements.

Kind regards,

Elena G. Tolkacheva, PhD

Academic Editor

PLOS ONE

Additional Editor Comments (optional):

Please correct minor points indicated by the Reviewer (below)

Reviewers' comments:

Reviewer's Responses to Questions

**Comments to the Author**

1. If the authors have adequately addressed your comments raised in a previous round of review and you feel that this manuscript is now acceptable for publication, you may indicate that here to bypass the “Comments to the Author” section, enter your conflict of interest statement in the “Confidential to Editor” section, and submit your "Accept" recommendation.

Reviewer #2: (No Response)

2. Is the manuscript technically sound, and do the data support the conclusions?

Reviewer #2: Yes

3. Has the statistical analysis been performed appropriately and rigorously? 

Reviewer #2: I Don't Know

4. Have the authors made all data underlying the findings in their manuscript fully available?

Reviewer #2: No

5. Is the manuscript presented in an intelligible fashion and written in standard English?

Reviewer #2: Yes

6. Review Comments to the Author

Reviewer #2: Thank you for making the corrections. In the process of looking over your manuscript, I happened to notice a few minor typos that you may want to correct, but I did not read everything in detail this time.

1) It looks like there is a typo in the Table 2 caption (“Nodel”).

2) On p. 8 you presumably meant “stimulation current Iapp” and not “simulation current Istim” (missing a ‘t’ in simulation).

3) I think there are some X’s that should have been capitalized in Eqs 17-18.

Thank you for your attention.

7. PLOS authors have the option to publish the peer review history of their article (what does this mean?). If published, this will include your full peer review and any attached files.

Reviewer #2: No

---

## [Editor Report · Acceptance letter]

1 Apr 2021

PONE-D-20-04786R3 

Parameter variations in personalized electrophysiological models of human heart ventricles  

Dear Dr. Ushenin:

I'm pleased to inform you that your manuscript has been deemed suitable for publication in PLOS ONE. Congratulations! Your manuscript is now with our production department. 

Kind regards, 

on behalf of

Dr. Elena G. Tolkacheva 

Academic Editor

PLOS ONE